# Excitatory projections from the nucleus reuniens to the medial prefrontal cortex modulate pain and depression-like behaviors in mice

**Shu-Ting Bao[1], Fang Rao[1], Cui Yin[1,2,3], Yong Niu[4], Jun-Li Cao[1,2,3,5]\*, Cheng Xiao[1,2,3]\*, Chunyi Zhou[1,2,3]\***

**1** Jiangsu Province Key Laboratory of Anesthesiology, School of Anesthesiology, Xuzhou Medical University, Xuzhou, China, **2** Jiangsu Province Key Laboratory of Anesthesia and Analgesia Application Technology, Xuzhou Medical University, Xuzhou, China, **3** NMPA Key Laboratory for Research and Evaluation of Narcotic and Psychotropic Drugs, School of Anesthesiology, Xuzhou Medical University, Xuzhou, China, **4** Key Laboratory of Chemical Safety and Health, National Institute for Occupational Health and Poison Control, Chinese Center for Disease Control and Prevention, Beijing, China, **5** Department of Anesthesia, Affiliated Hospital of Xuzhou Medical University, Xuzhou, China

\* Cao0310@aliyun.com (J-LC); xchengxj@xzhmu.edu.cn (CX); chunyi.zhou@xzhmu.edu.cn (CZ)

## Abstract

The medial prefrontal cortex (mPFC) is implicated in emotional processing, cognition, and pain sensation, moreover, its circuitry undergoes neuroplastic changes in chronic pain. Although the nucleus reuniens (RE) of the thalamus provides significant glutamatergic inputs to the mPFC, it remains unclear whether this projection contributes to plasticity changes in the mPFC and pain-related behaviors in chronic pain. Using fiber photometry, we demonstrated that RE neurons responded to pain stimulation and emotional changes. Optogenetic activation of RE neurons and their projections to the mPFC (RE-mPFC projection) elicits hyperalgesia and depression-like behaviors in naïve mice. In a neuropathic pain mouse model, RE neurons were hyperactive, and the RE-mPFC projection was enhanced with a marked preference for the part innervating GABAergic circuits in the mPFC to that controlling mPFC neurons projecting to the ventrolateral periaqueductal gray (vlPAG). Expectedly, optogenetic inhibition of RE neurons and the RE-mPFC projection ameliorated pain-like and depression-like behaviors in neuropathic pain mice. Additionally, chemogenetic inhibition of RE-mPFC neurons conferred analgesia in neuropathic pain mice exposed to both acute and chronic morphine. Our findings highlight the significant role of the RE-mPFC pathway in neuropathic pain comorbid with depression, suggesting its potential as a target for treatment of neuropathic pain.

**Data availability statement:** The source data in this study are included in S1 Data submitted as a part of Supporting information.

**Funding:** This work was supported by the Sci-Tech Innovation 2030-Major Project (https://service.most.gov.cn/index/) (2021ZD0203100 to JLC), the National Natural Science Foundation of China (https://www.nsfc.gov.cn/) (82371242 to CZ; 82171235 to CZ; 82071231 to CX; 82271293 to CX), the Fund for Jiangsu Province Specially-Appointed Professor (to CX and CZ), the Natural Science Research of Jiangsu Higher Education Institutions of China (https://info.jse.edu.cn) (23KJA320006 to CZ; 23KJA320007 to CX), the Leadership Program in Xuzhou Medical University (https://www.xzhmu.edu.cn/) (JBGS202203 to CX), and the Postgraduate Research & Practice Innovation Program of Jiangsu Province (KYCX23_2955 to STB). The funders had no role in study design, data collection and analysis, decision to publish, or preparation of the manuscript.

**Competing interests:** The authors have declared that no competing interests exist.

**Abbreviations:** AAV, adeno-associated virus; BIC, bicuculline; CNO, clozapine-N-oxide; CNQX, 6-Cyano-7-nitro-quinoxaline-2, 3-dione disodium salt hydrate; CPA, conditioned place aversion; CPP, conditioned place preference; EPM, elevated plus maze; FST, forced swim test; Glu, glutamatergic; mPFC, medial prefrontal cortex; PBS, phosphate-buffered saline; PFA, paraformaldehyde; PWL, paw withdrawal latency; PWT, paw withdrawal threshold; RE, nucleus reuniens; RV, rabies virus; SD, standard deviation; SNI, spared nerve injury; TST, tail suspension test; TTX, tetrodotoxin; VTA, ventral tegmental area; 4-AP, 4-aminopyridine.

## Introduction

Chronic pain is one of the important symptoms in various diseases and affects up to 20% of the global population [1]. Patients with chronic pain often experience comorbidities such as depression and impaired memory, which have not been adequately treated by medication [1–3]. This presents a significant clinical challenge and an urgent need to explore the cellular and circuit basis for the comorbidity of pain and depression.

The development of chronic pain is linked to structural and functional changes in neural circuits that are involved in sensory and affective functions [4,5]. This may provide a pathophysiological basis for the comorbidity of pain and depression in chronic pain. The medial prefrontal cortex (mPFC), including the anterior cingulate cortex, the prelimbic, and the infralimbic cortex, is a region that is significantly affected in chronic pain [6–9] and may play a crucial role in this comorbidity. The mPFC is activated by painful stimuli and shows reduced activity in chronic pain states [9,10]. In patients with chronic pain, the PFC undergoes a decrease in gray matter volume and a reduction in connectivity with other cortical regions [11–13]. Conversely, transcranial stimulation of the mPFC has been shown to improve hyperalgesia, depression, and working memory deficits in patients with chronic pain [14]. Similarly, in rodent models, prefrontal pyramidal neurons exhibited decreased excitability under peripheral nerve injury-induced neuropathic pain conditions [8,15,16]. Notably, manipulations that selectively stimulate PFC pyramidal neurons ameliorate ongoing chronic pain and pain-related emotional dysfunctions [6,8,16–18]. While it is known that the local circuitry within the mPFC is finely tuned by synaptic inputs from multiple brain regions [8,14,19,20], it remains uncertain which upstream nuclei recruit or drive specific pathophysiological changes in the mPFC in the context of chronic pain.

The nucleus reuniens (RE) of the thalamus provides strong glutamatergic (Glu) synaptic inputs to the mPFC and hippocampus, while also receives Glu inputs from the mPFC [21,22]. The RE supports the communication between the mPFC and hippocampus and facilitates working memory and executive functions [22,23]. Beyond memory, the RE plays a role in resilience to chronic stress-induced depression [24]. However, it has not been reported how RE neurons modulate the mPFC circuit, which consists of Glu projection neurons and GABAergic interneurons. Moreover, it remains unclear whether the RE-mPFC projection is modified and implicated in hyperalgesia and depression-like behaviors in chronic pain.

In this study, we investigated the contribution of RE neurons and their projections to the mPFC (RE-mPFC) to hyperalgesia and depression-like behavior in chronic pain. We demonstrated that RE neurons not only responded to acute mechanical, thermal, and emotional stimuli but also modulated pain thresholds and depression-like behaviors. Furthermore, we revealed modifications in the RE-mPFC pathway in neuropathic pain and confirmed that reversing these modifications mitigates hyperalgesia and depression-like behaviors in mice with neuropathic pain. Our findings underscore the involvement of the RE-mPFC pathway in chronic pain and comorbid depression, highlighting it as a potential target for effective treatment.

## Results

### RE neurons are activated by noxious stimuli and emotional changes

We labeled RE Glu neurons with a recombinant adeno-associated virus (AAV) vector under the control of CaMKII promoter (AAV-CaMKII-eYFP) (S1A and S1B Fig). In these mice, repetitive stimulation of hind paws with von Frey filament (2 g) significantly increased c-Fos(+) Glu neurons in the RE (S1C–S1E Fig). This result is consistent with previous reports [25,26]. We transfected a genetically encoded calcium sensor (GCaMP6s) or eYFP into RE Glu neurons by injecting AAV-CaMKII-GCaMP6s or AAV-CaMKII-eYFP into the RE and implanted an optic fiber into the injection site (Fig 1A and 1B). These mice then underwent either spared nerve injury (SNI) or sham surgery. Three weeks after viral injection, we performed real-time in vivo fiber photometry recording from RE Glu neurons. We observed that in sham mice, GCaMP6 signals in RE Glu neurons increased significantly in response to stimulation on the hind paw with a 2.0 g von Frey filament but not with a 0.4 g von Frey filament (Fig 1C–1E); in contrast, SNI mice showed a robust response to the stimulation on the hind paw with the 0.4 g von Frey filament (Fig 1C–1E). The data are consistent with a lower mechanical paw withdrawal threshold (PWT) in SNI mice than sham mice (S2A Fig). RE Glu neurons labeled with eYFP did not respond to 2 g von Frey filament stimulation (Fig 1C–1E). Additionally, thermal stimulation with a 48°C heating block on the hind paw of SNI mice induced an equivalent peak GCaMP6 response but with a shorter latency compared to sham mice (Fig 1F–1H). This is consistent with the observation that SNI mice exhibited a shorter hind paw withdrawal latency (PWL) to the 48°C heating block than sham mice (S2B Fig). GCaMP6 signals did not change when mice lifted their feet during walking (Fig 1F—1H). These data indicate that RE Glu neurons are activated by both mechanical and thermal stimulation in both physiological and neuropathic pain conditions.

We further examined how the activity of RE Glu neurons is modulated in response to stimuli linked to changes in emotional states. GCaMP6 signals in RE neurons increased when mice entered into the open arms in an elevated plus maze (EPM) and decreased when they retreated into the closed arms, with stronger responses observed in SNI mice compared to naïve mice (Fig 1I−1N). In addition, a 1 s air puff (aversive stimulation) onto the snout significantly increased GCaMP6 signals in both naïve and SNI mice, but the response was stronger in SNI mice (Fig 1O−1Q). In contrast, interaction with an unfamiliar juvenile mouse of the same sex elicited comparable GCaMP6 signal changes in both groups (Fig 1R−1T). These data suggest that RE Glu neurons may be involved in the processing of pain, aversion, and anxiety-related stimuli, with enhanced responses in SNI mice.

### Activation of RE Glu neurons induces hyperalgesia and negative emotion

Following our observation that the activity of RE neurons was elevated in response to various external stimuli, we sought to determine whether RE Glu neurons modulate pain and depression-like behaviors in naïve mice.

To address this issue, we injected AAV-CaMKII-ChR2-eYFP or AAV-CaMKII-eYFP (as control) into the RE of mice (ChR2 or eYFP mice) and implanted an optical fiber above the injection site (Fig 2A and 2B). Patch-clamp recordings confirmed that blue light stimulation (472 nm, 10 ms, 20 Hz, 2 mW) evoked inward currents and firing in ChR2-expressing RE Glu neurons (Fig 2C). To ascertain the role of RE Glu neurons in pain perception, we assessed mechanical PWT and thermal PWL of ChR2- and eYFP-expressing mice in the presence or the absence of blue light stimulation of the RE. Activation of RE Glu neurons with blue light (472 nm, 5 ms, 20 Hz, 5 mW) significantly reduced mechanical PWT and thermal PWL on hind paws in naïve ChR2 mice (Fig 2D–2G), but not in eYFP mice (Fig 2H and 2I). These data suggest that stimulating RE Glu neurons induces pain-like behaviors in naïve mice.

In addition to pain-like behaviors, we examined whether RE Glu neurons regulate emotional behaviors in mice. We employed a conditioned place preference or aversion (CPP or CPA) paradigm as shown in Fig 2J to determine whether repetitive activation of RE Glu neurons leads to aversion. Over a 3 days' conditioning period, both ChR2- and eYFP- mice received blue light stimulation (20 Hz, 5 ms, 4 mW) paired with one of the two chambers in the afternoon, whereas no

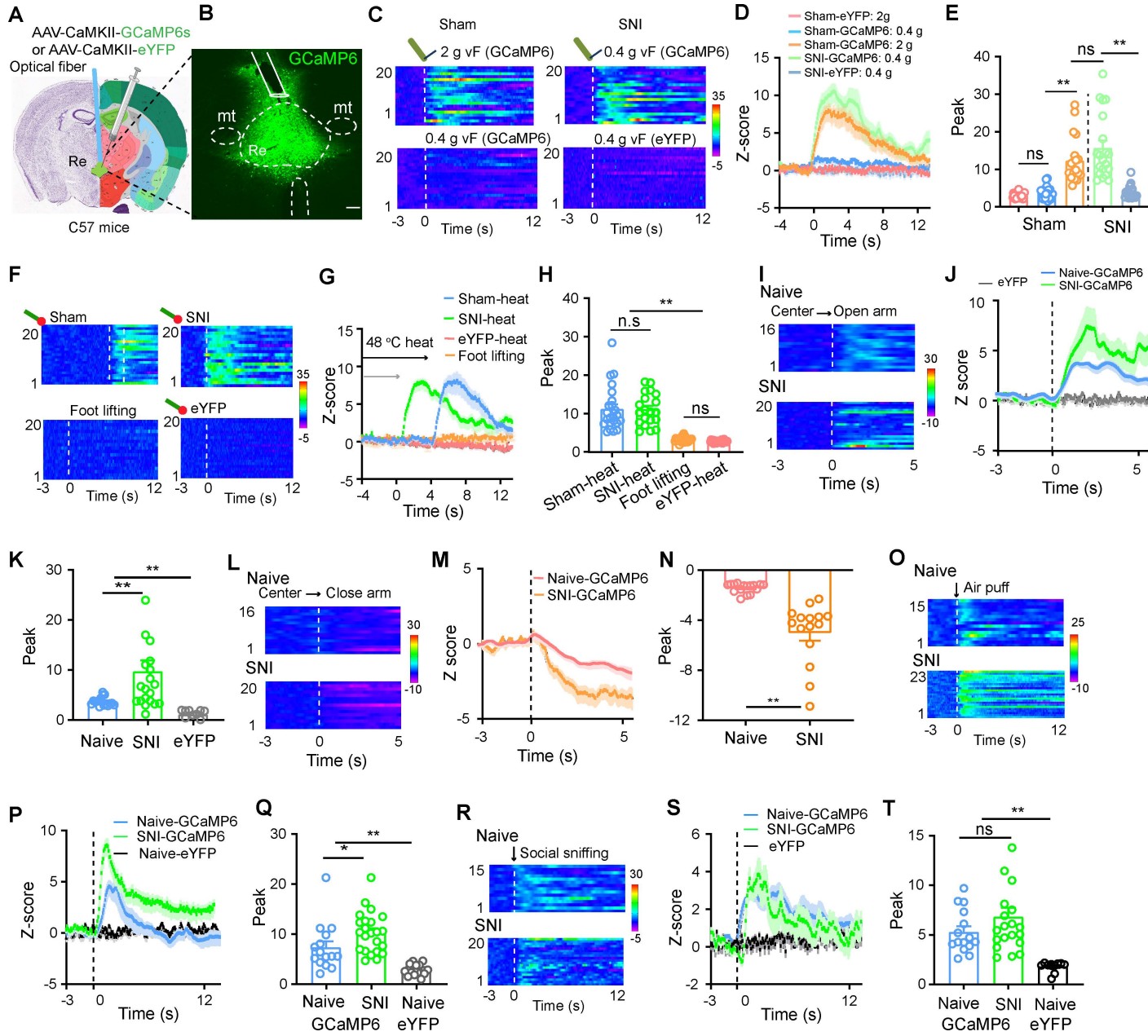

**Fig 1. RE neurons respond to pain-like stimuli, emotional changes, and social interaction in SNI mice. (A, B)** Diagram of virus injection (A) and an example image (B) of GCaMP6s expression in the RE. (A) Nissl (left) and anatomical annotations (right) from the Allen Mouse Brain Atlas (https:// mouse.brain-map.org) and Allen Reference Atlas-Mouse Brain (https://atlas.brain-map.org). **(C–E)** Heat maps (C), averaged normalized traces (D), and summary (E) of changes in GCaMP6s and eYFP signals in the RE in response to von Frey filament (vF) stimulation on the injured hind paw in SNI or sham mice ($n = 20$ trials from 6 mice in each group). $F_{(4, 95)} = 29.58$, $P < 0.0001$. **(F–H)** Heat maps (F), averaged normalized traces (G), and summary (H) of changes in GCaMP6s and eYFP signals in the RE in response to thermal stimulation or random foot lifting on the injured hind paw in SNI or sham mice (H, $F_{(3, 82)} = 36.9$, $P < 0.0001$) ($n = 20 - 23$ trials from 6 mice in each group). **(I–K)** Heat maps (I), averaged normalized traces (J), and summary data (K, $F_{(2, 43)} = 6.74$, $P = 0.003$, $n = 16 - 20$ trials from 6 mice in each group) showing changes in GCaMP6s and eYFP signals in naïve and SNI mice during exploration of the open arms in the elevated plus maze (EPM). **(L–N)** Heat maps (L), averaged normalized traces (M), and summary data (N, $t = 5.78$, $P < 0.0001$, $n = 16 - 20$ trials from 6 mice in each group) showing changes in GCaMP6s signals in naïve and SNI mice during exploration of the close arms in EPM. **(O–Q)** Heat maps (O), averaged normalized traces (P), and summary data (Q, $F_{(2, 54)} = 22.68$, $P < 0.0001$, $n = 15 - 23$ trials from 6 mice in each group) showing changes in GCaMP6s and eYFP signals in naïve and SNI mice in response to 1 s air puff directed toward the face. **(R–T)** Heat maps (R), averaged normalized traces (S), and summary data (T, $F_{(2, 44)} = 10.74$, $P = 0.0002$, $n = 15 - 20$ trials from 6 mice in each group) showing

changes in GCaMP6s and eYFP signals in naïve and SNI mice during sniffing behavior towards a novel conspecific mouse. Dashed lines in (C, F, O, R) indicate stimulus onset. Dashed lines in (I, L) indicate walking into the open or closed arms. vF: von Frey filament. Scale bar in (B), 100 μm.*$P < 0.05$, **$P < 0.01$; ns not significant. One-way ANOVAs with Tukey's post-hoc analysis for (E, H, K, Q, T). Two-tailed unpaired $t$ test for (N). $n = 6$ mice in each group. Data are available in S1 Data as a part of Supporting information.

stimulation was applied in the opposing chamber in the morning. During the test session, we observed that ChR2 mice spent less time in the blue-light-conditioned chamber relative to the precondition session, with no changes in velocity in the paired chamber (Fig 2K–2M). These results suggest that stimulating RE Glu neurons may cause aversion. We observed that optogenetic activation of RE Glu neurons modulated depression-like behaviors. Specifically, it reduced the time spent sniffing body of novel conspecific mice (Fig 2N−2P), suggesting decreased social motivation, a putative symptom of depression. Additionally, it increased immobility time in both the tail suspension test (TST) and the forced swim test (FST) (Fig 2Q and 2R), which are indicators of helplessness associated with depression. These data suggest that stimulation of RE Glu neurons may lead to depression-like states.

### Inhibition of RE neurons alleviates neuropathic pain-associated behaviors

We next established a neuropathic pain mouse model with SNI. Consistent with our recent studies [27,28], mechanical and thermal hyperalgesia on the hind paw appeared shortly after SNI surgery and persisted over time (S2A and S2B Fig). Additionally, SNI mice showed increased immobility time in both TST and FST (S2C and S2D Fig), as well as impaired social motivation, evidenced by reduced time spent investigating novel mice 4 weeks after SNI (S2E and S2F Fig). These results suggest that SNI mice exhibit hypersensitivity in pain sensation and display depression-like behaviors.

As stimulation of RE Glu neurons induced pain behaviors (Fig 2), we investigated their involvement in the pathophysiology of SNI mice. To address this question, we injected AAV-CaMKII-NpHR-eYFP or AAV-CaMKII-eYFP into the RE (Fig 3A). Following SNI or sham surgery, we observed that eYFP-labeled RE neurons in SNI mice exhibited enhanced excitability relative to those in sham mice (Fig 3B and 3C). In ChR2 mice subjected to SNI surgery, optogenetic activation of RE Glu neurons did not further reduce pain thresholds, as mechanical and thermal hypersensitivity remained unchanged (S3A and S3B Fig).

After confirming yellow light (589 nm, 0.5 s, 2 mW) sufficiently silenced NpHR-expressing RE Glu neurons 3 weeks after virus injection (Fig 2D), we examined whether reversing hyperexcitability of RE Glu neurons with optogenetic inhibition mitigates pain-related behaviors in SNI mice. Although optogenetic inhibition of RE Glu neurons did not alter mechanical PWT and thermal PWL in naïve mice (S3C–S3F Fig), it induced significant increases in both mechanical PWT and thermal PWL on the hind paw of SNI mice (Fig 3E—3H). In addition to the pain-relieving effect in SNI-NpHR mice, optogenetic inhibition established CPP with no changes in velocity in the paired chamber (Fig 3I−3K), increased time investigating novel mice (Fig 3L and 3M), and shortened immobility time in the TST and FST (Fig 3N and 3O), compared to SNI-eYFP mice.

We also examined whether inhibition of RE neurons mitigates pain hypersensitivity and negative emotions in female SNI mice. We injected AAV-CaMKII-NpHR-eYFP or AAV-CaMKII-eYFP and implanted an optical fiber into the RE in the right hemisphere, and conducted SNI or sham surgery on the right side (S3G Fig). We observed that optogenetic inhibition of RE neurons significantly increased mechanical and thermal thresholds on the right hind paw (S3H and S3I Fig), induced CPP (S3J–S3K Fig), improved social interaction (S3M and S3N Fig), and reduced immobility time in the TST and FST (S3O and S3P Fig).

These results demonstrate that reversing hyperactivity in RE Glu neurons in male and female SNI mice is sufficient to mitigate nerve injury-induced pain-like behaviors and comorbid depression-like behaviors.

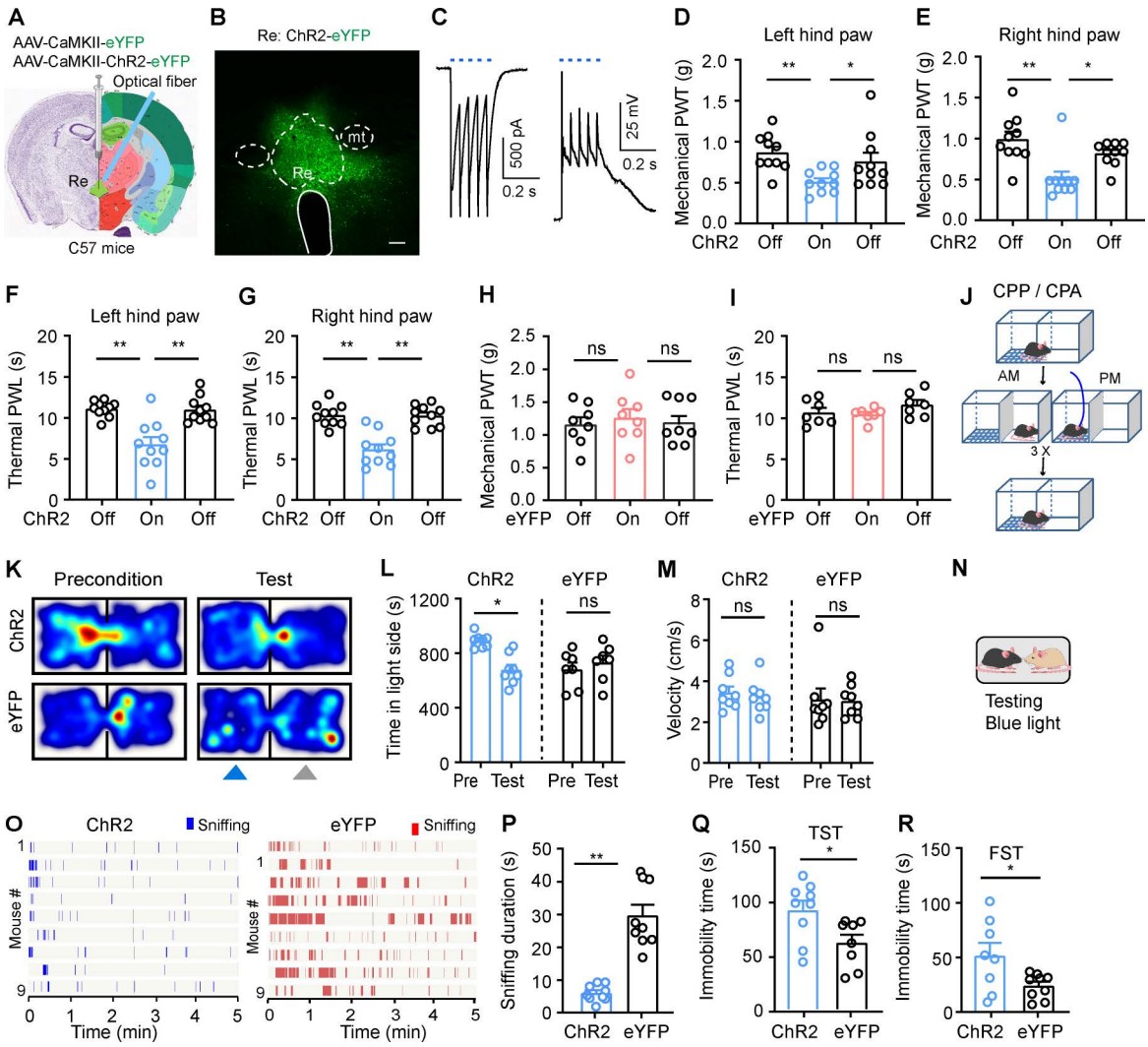

**Fig 2. Optogenetic activation of RE neurons induces pain-like hypersensitivity, aversion, and depression-like behaviors. (A)** Schematic diagram of optogenetic activation of RE neurons in naïve mice. Nissl (left) and anatomical annotations (right) from the Allen Mouse Brain Atlas (https://mouse.brain-map.org) and Allen Reference Atlas-Mouse Brain (https://atlas.brain-map.org). **(B)** ChR2 expression in the RE. **(C)** Currents (left) and action potentials (right) induced by photostimulation (473 nm, 5 ms, blue bars) in ChR2-eYFP(+) RE neurons. **(D, E)** Effect of optogenetic activation of RE neurons on mechanical paw withdrawal threshold (PWT) (D, $F_{(2, 27)}$ = 5.29, $P = 0.01$; E, $F_{(2, 27)}$ = 9.88, $P = 0.0006$; $n = 10$ each group) and thermal paw withdrawal latency (PWL) (F, $F_{(2, 27)}$ = 15.58, $P < 0.0001$; G, $F_{(2, 27)}$ = 23.15, $P < 0.0001$; $n = 10$ mice in each group) on both hind paws. **(H, I)** Effect of blue light illumination in eYFP-expressing RE neurons on mechanical PWT (H, $F_{(2, 21)}$ = 0.17, $P = 0.84$; $n = 8$ in each group) and thermal PWL (I, $F_{(2, 18)}$ = 1.57, $P = 0.23$; $n = 7$ mice in each group) on hind paws. **(J–L)** Timeline for the place conditioning test (J). Example heat maps (K), time spent (L) and velocity **(M)** in the blue-light-paired chamber during the preconditioning (Pre) (Day 1) and test (Day 5) sessions in ChR2 mice and eYFP mice (L, $F_{(1, 18)}$ = 13.26, $P = 0.0019$; M, $F_{(1, 7)}$ = 0.40, $P = 0.55$; $n = 8$ ChR2 mice, $n = 9$ eYFP mice). **(N)** Schematic diagram of social motivation test. **(O, P)** Raster plot showing sniffing episodes (O) and total time that eYFP and ChR2 mice spent for sniffing the novel stimulus mice (P, $t = 7.01$, $P < 0.0001$) during blue light illumination of the RE. **(Q, R)** Immobility time of ChR2 mice and eYFP mice during blue light illumination into the RE in the tail suspension test (TST) (Q, $t = 2.49$, $P = 0.02$) and the forced swim test (FST) (R, $t = 2.38$, $P = 0.03$). $n = 8$ ChR2 mice, $n = 9$ eYFP mice. Blue light (473 nm, 20 Hz, 5 ms pulse width, 4 mW) is applied into the RE in behavioral experiments. Scale bars: 100 μm. *$P < 0.05$, **$P < 0.01$, ns not significant. One-way ANOVAs with Tukey's post-hoc analysis for (D−I). Two-way ANOVAs with Tukey's post-hoc analysis for (L, M). Two-tailed unpaired $t$-tests for (P−R). Data are available in S1 Data as a part of Supporting information.

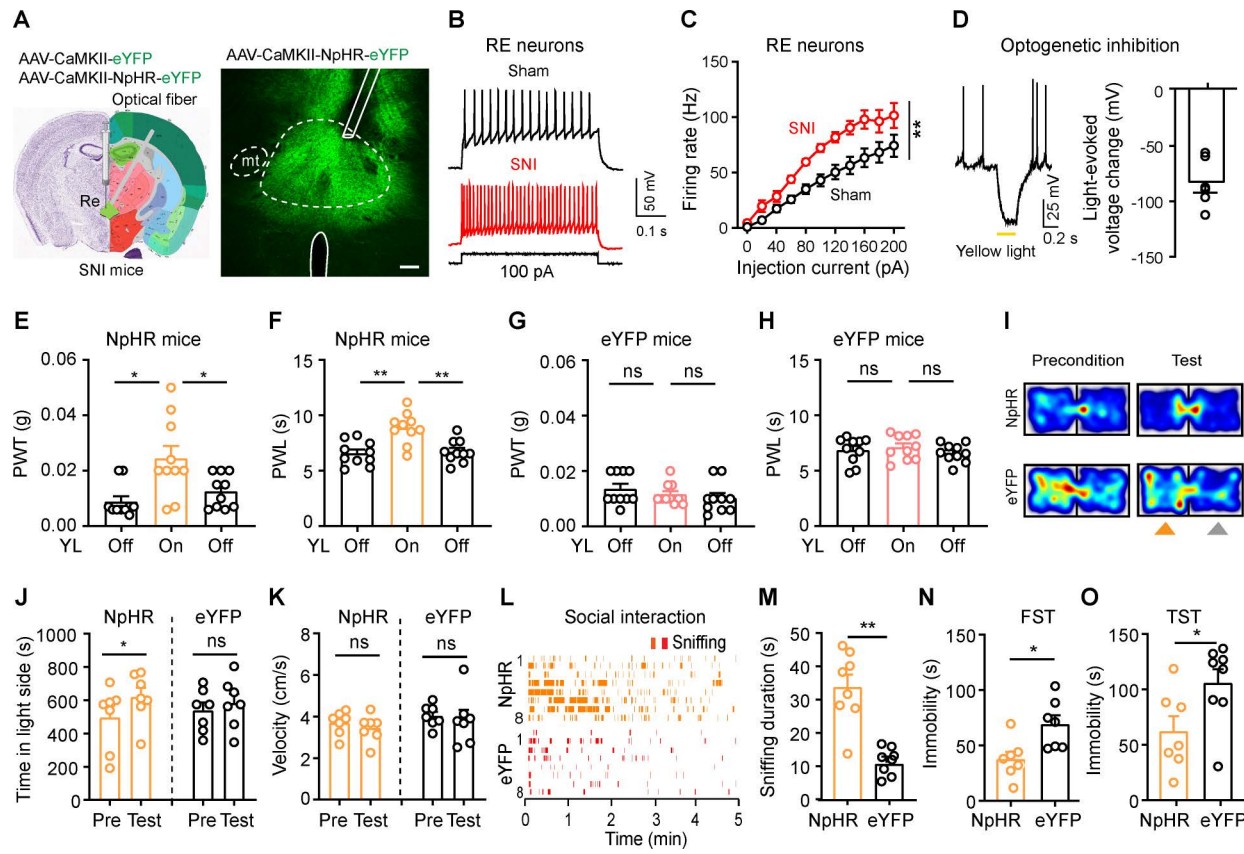

**Fig 3. Inhibition of RE neurons improves hyperalgesia and emotions in SNI mice. (A)** Schematic diagram of optogenetic inhibition of RE neurons in SNI mice. Nissl (left) and anatomical annotations (right) from the Allen Mouse Brain Atlas (https://mouse.brain-map.org) and Allen Reference Atlas-Mouse Brain (https://atlas.brain-map.org). **(B, C)** Example traces of depolarizing current injection-induced firing (B) and summary of firing frequencies (C, $F_{(1, 14)}$ = 10.67, $P = 0.006$) in eYFP-labeled RE neurons in sham ($n = 9$ neurons) and SNI ($n = 6$ neurons) mice. **(D)** Example trace and quantification of yellow light-induced hyper-polarization in NpHR-expressing RE neurons ($n = 5$ neurons). **(E, F)** Effect of yellow light (YL) illumination of RE neurons on mechanical PWT ($F_{(2, 27)}$ = 7.28, $P = 0.003$) and thermal PWL ($F_{(2, 27)}$ = 11.93, $P = 0.0002$) in NpHR mice ($n = 10$ mice). **(G, H)** Effect of yellow light (YL) illumination of RE neurons on mechanical PWT ($F_{(2, 27)}$ = 1.01, $P = 0.36$) and thermal PWL ($F_{(2, 27)}$ = 0.59, $P = 0.56$) in eYFP mice ($n = 10$ mice). **(I–K)** Example heat maps (I), time spent (J, Time, $F_{(1, 12)}$ = 13.24, $P = 0.003$) and velocity (K, $F_{(1, 12)}$ = 1.31, $P = 0.27$) in the yellow-light-paired chamber during the preconditioning (Day 1) and test (Day 5) sessions for NpHR mice ($n = 7$) and eYFP mice ($n = 7$). **(L, M)** Raster plot showing sniffing episode (L) and total time spent sniffing (M, $t = 5.72$, $P < 0.0001$) in NpHR ($n = 8$) and eYFP ($n = 8$) mice tested during yellow light illumination of the RE. **(N, O)** Immobility time in the FST (N, $t = 2.37$, $P = 0.03$) and TST (O, $t = 2.94$, $P = 0.01$) in NpHR ($n = 7$) and eYFP ($n = 8$) mice during yellow light illumination of RE neurons.

## RE Glu neurons anatomically and functionally connect to both vlPAG-projecting neurons and GABAergic interneurons in the mPFC

Given that the mPFC is associated with behaviors related to pain and emotions [6], we next focused on the RE-mPFC pathway. We first asked whether RE inputs specifically target pyramidal projection neurons or GABAergic interneurons within the mPFC. Using a cell-type specific retrograde transsynaptic tracing with modified rabies virus and AAV vectors (RV-EnvA-ΔG-dsRed, AAV-EF1α-DIO-RVG and AAV-EF1α-DIO-TVA-eGFP) and Cre-dependent mouse lines (Vgat-Cre and CaMKII-Cre), we confirmed that the RE was one of upstream nuclei of mPFC GABAergic and Glu neurons (S4A–S4J Fig). We also employed a viral vector (AAV-CaMKII-ChR2-eYFP) to show that RE Glu neuron-originating eYFP-labeled fibers and terminals in the mPFC were mainly located in layers I, IV and V (S4K–S4N Fig). Given that the PAG, a key midbrain structure responsible for descending inhibition of ascending nociceptive inputs [29], is a major subcortical target

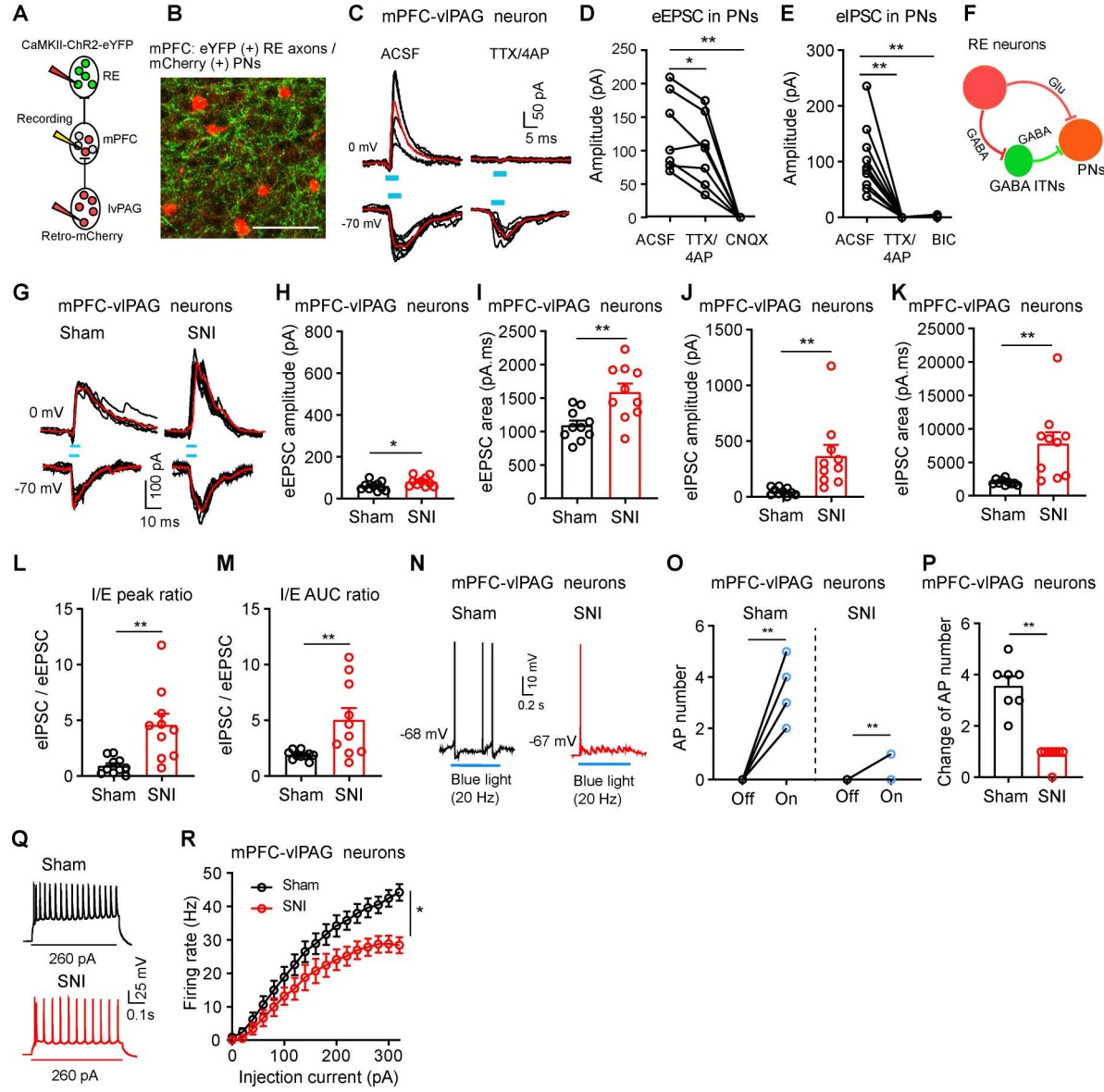

**Fig 4. Dysfunction of the RE projection into vlPAG-projecting mPFC neurons in SNI mice. (A)** Diagram illustrating patch-clamp recordings on vlPAG-projecting mPFC neurons in response to activation of ChR2 in RE terminals. **(B)** ChR2 and mCherry expression in the mPFC. **(C)** Light-evoked EPSC (−70 mV) and IPSC (0 mV) recorded from mCherry(+)-mPFC-vlPAG neurons before and during perfusion of TTX (0.5 μM) and 4-AP (100 μM). **(D, E)** Amplitude of light-evoked EPSCs (D) and IPSCs (E) on mPFC-vlPAG neurons (D, $F_{(1.2, 7.3)}$ = 28.73, $P < 0.0007$, $n = 7$ cells from 4 mice; E, $F_{(1.0, 7.0)}$ = 29.57, $P = 0.0004$, $n = 10$ cells from 4 mice). **(F)** Diagram showing that RE neurons affect mPFC-vlPAG neurons through direct and indirect synaptic transmission. **(G)** Light-evoked EPSC (−70 mV) and IPSC (0 mV) on mPFC-vlPAG neurons from sham and SNI mice. **(H−K)** Amplitude and area of light-evoked EPSCs and IPSCs in mPFC-vlPAG neurons of sham and SNI mice (H, $t = 2.19$, $P = 0.04$; I, $t = 3.38$, $P = 0.003$; J, $t = 3.16$, $P = 0.005$; K, $t = 3.16$, $P = 0.005$; $n = 10$ cells from 4 mice in each group). **(L, M)** Ratios of light-evoked IPSCs to EPSCs (I/E ratio) in sham and SNI mice (L, $t = 3.51$, $P = 0.003$; M, $t = 2.96$, $P = 0.008$; $n = 10$ cells). **(N–P)** Representative traces (N), dot plots (O), and summary (P) showing 1 s 20 Hz blue light (5 ms, 2 mW)-evoked action potentials in mPFC-vlPAG neurons in sham and SNI mice (O, $t = 9.68$, $P < 0.0001$ for sham; $t = 7$, $P = 0.0002$ for SNI; P, $t = 7.32$, $P < 0.0001$; $n = 7 − 8$ cells from 4 mice in each group). **(Q)** Example traces of firing evoked by 260 pA current injection recorded from mPFC-vlPAG neurons from sham and SNI mice, respectively. **(R)** Summary of the frequencies of firing evoked by depolarizing current injection in mPFC-vlPAG neurons of SNI and sham mice ($F_{(16, 336)}$ = 5.27, $P < 0.0001$, $n = 14$ cells from 4 sham mice, $n = 11$ cells from 4 SNI mice). Scale bar: 100 μm. *$P < 0.05$, **$P < 0.01$; One-way repeated measures ANOVAs with Tukey's post-hoc analysis for (D, E); Two-tailed unpaired $t$-tests for (H−M, P); Two-tailed paired $t$ test for (O); Two-way repeated measures ANOVA with Tukey's post-hoc analysis for (R). Data are available in S1 Data as a part of Supporting information.

of mPFC projection neurons in the layer V [30], we injected AAV retro-hSyn-mCherry into the ventrolateral periaqueductal gray (vlPAG) to label vlPAG-projecting mPFC (mPFC-vlPAG) neurons with mCherry (Fig 4A). As illustrated in Fig 4B, mCherry-labeled mPFC-vlPAG neurons in the layer 5 of the mPFC were surrounded by eYFP-labeled fibers.

Using whole-cell patch-clamp technique, we observed that blue light stimulation of ChR2-labeled RE axons in the mPFC elicited EPSCs at a holding potential of −70 mV and IPSCs at a holding potential of 0 mV in mPFC-vlPAG neurons (Fig 4C); the light-evoked EPSCs in these neurons were not dramatically diminished in the presence of 0.5 µM tetrodotoxin (TTX) and 0.1 mM 4-aminopyridine (4-AP), but were completely blocked by AMPA receptor antagonist (20 µM CNOX) (Fig 4D); the light-evoked IPSCs were eliminated by TTX and 4-AP or 10 µM bicuculline (BIC) (Fig 4E). These results suggest that RE neurons innervate mPFC-vlPAG neurons through both polysynaptic (indirectly via mPFC GABAergic interneurons) and monosynaptic (directly) connections (Fig 4F). To test whether RE Glu neurons innervate mPFC GABAergic neurons, we injected AAV-DIO-eYFP into the mPFC of Vgat-Cre mice, and AAV-CaMKII-ChR2 into the RE and AAV retro-hSyn-mCherry into the vlPAG to label mPFC-vlPAG neurons. We recorded from eYFP-labeled GABAergic neurons in the mPFC, which were not co-labeled with mCherry (S5A Fig). These GABAergic neurons exhibited blue light-evoked, TTX+4AP-resistant EPSCs (S5C and S5D Fig). Thus, RE neurons may regulate mPFC-vlPAG neurons by providing excitatory monosynaptic inputs directly and inhibitory polysynaptic inputs via GABAergic interneurons, maintaining balanced activity of mPFC-vlPAG neurons.

In addition, we examined whether mPFC-projecting RE neurons also send collaterals to the hippocampus. We injected retrograde viral vectors: AAV retro-hSyn-eYFP and AAV retro-hSyn-mCherry into the ventral hippocampus and mPFC (S6A and S6B Fig), respectively. However, RE neurons co-labeled with both eYFP and mCherry accounted for less than 15% of RE neurons labeled by eYFP or mCherry (S6C and S6D Fig). These data indicate that most of RE neurons projecting to the mPFC or the ventral hippocampus do not have collateral projection to the other ones.

## The RE-mPFC circuit is modified in SNI mice

We next asked whether RE inputs to mPFC-vlPAG neurons are altered in SNI mice. We used the same strategy as illustrated in Fig 4A to transfect RE neurons and their processes with ChR2-eYFP and label mPFC-vlPAG neurons with mCherry (Fig 4A and 4B). We recorded the peak amplitudes and the area of blue light stimulation-evoked EPSCs and IPSCs in mPFC-vlPAG neurons to quantify the synaptic currents. We found that blue light stimulation-evoked EPSCs and IPSCs in mPFC-vlPAG neurons from SNI mice (4 weeks after surgery) showed larger amplitudes and greater current areas compared to those from sham control mice (Fig 4G—4K). The I:E ratios calculated using both the peak amplitudes and current area were significantly greater in mPFC-vlPAG neurons from SNI mice compared to sham controls (Fig 4L and 4M). These data suggest that GABAergic inputs to these neurons are enhanced to an extent much stronger than excitatory inputs following SNI. Our observation that the RE-mPFC projection onto GABAergic neurons was enhanced dramatically (S5E and S5F Fig) confers mPFC GABAergic neurons as important sources for the enhanced inhibitory inputs to mPFC-vlPAG neurons. Consistent with this, we recorded firing from mPFC-vlPAG neurons and observed that blue light pulses induced less action potentials in mPFC-vlPAG neurons in SNI mice than in sham mice (Fig 4N–4P). However, when a GABA receptor blocker (10 µM BIC) was applied, firing frequency in mPFC-vlPAG neurons during blue light stimulation of the RE-mPFC projection became similar between sham and SNI mice (S5G and S5H Fig). Therefore, in SNI mice, the monosynaptic excitatory effect of the RE-mPFC projection onto mPFC-vlPAG neurons may be compromised by enhancement of that onto mPFC GABAergic interneurons innervating mPFC-vlPAG neurons. We also observed another phenomenon: the frequencies of firing evoked by depolarizing current injections in mPFC-vlPAG neurons were significantly decreased in SNI mice relative to those in sham control mice (Fig 4N and 4Q); this difference was abolished in the presence of fast synaptic transmission blockers (20 µM 6-Cyano-7-nitro-quinoxaline-2, 3-dione disodium salt hydrate (CNQX) and 10µM BIC) (S5I and S5J Fig). These findings indicate that the reduced intrinsic excitability observed in SNI mice was largely due to altered synaptic input rather than changes in intrinsic excitability.

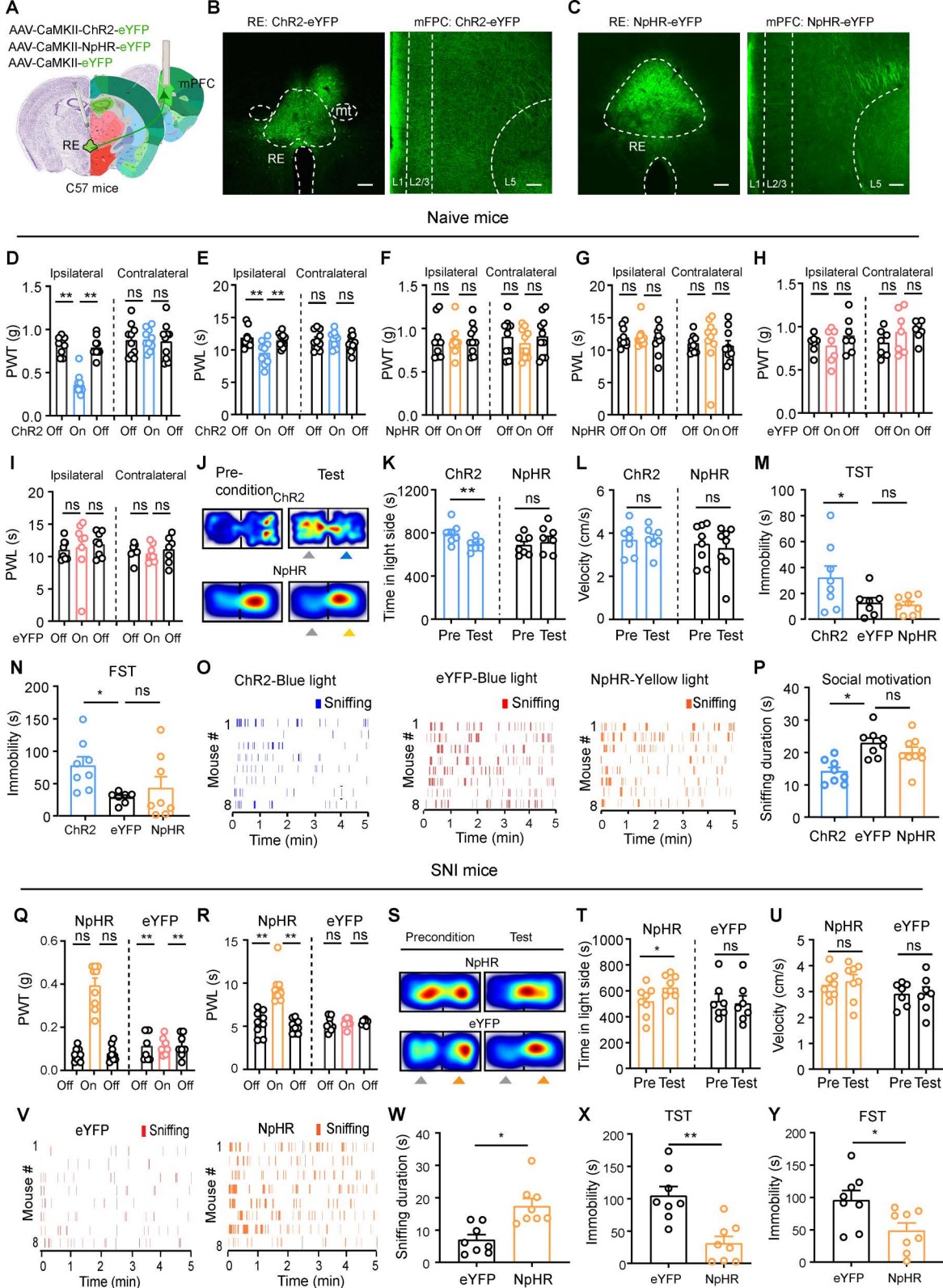

**Fig 5. The RE-mPFC projection modulates pain thresholds and depression-like behaviors in naïve mice. (A)** Schematic diagram of optogenetic modulation of the RE-mPFC projections. Nissl (left part of slices) and anatomical annotations (right part of slices) from the Allen Mouse Brain Atlas

([https://mouse.brain-map.org](https://mouse.brain-map.org)) and Allen Reference Atlas-Mouse Brain ([https://atlas.brain-map.org](https://atlas.brain-map.org)). **(B, C)** Example images of ChR2- and NpHR-expression pattern in the RE and mPFC. **(D, E)** Effect of unilateral blue light illumination of the RE-mPFC projection on mechanical PWT and thermal PWL in naïve ChR2 mice ($n = 10$). (D) PWT. Ipsilateral: $F_{(2, 27)} = 53.24$, $P < 0.0001$; Contralateral: $F_{(2, 27)} = 0.04$, $P = 0.96$. (E) PWL. Ipsilateral: $F_{(2, 27)} = 6.95$, $P = 0.004$; Contralateral: $F_{(2, 27)} = 52$, $P = 0.60$. **(F, G)** Effect of unilateral yellow light illumination of the RE-mPFC projection on mechanical PWT and thermal PWL in naïve NpHR mice ($n = 10$). (F) PWT. Ipsilateral: $F_{(2, 27)} = 0.05$, $P = 0.95$; Contralateral: $F_{(2, 27)} = 0.45$, $P = 0.64$. (G) PWL. Ipsilateral: $F_{(2, 27)} = 0.22$, $P = 0.81$; Contralateral: $F_{(2, 27)} = 2.4$, $P = 0.11$. **(H, I)** Effect of unilateral blue light illumination of the RE-mPFC projection on mechanical PWT and thermal PWL in naïve eYFP mice ($n = 7$). (H) PWT. Ipsilateral: $F_{(2, 18)} = 0.9$, $P = 0.42$; Contralateral: $F_{(2, 18)} = 1.33$, $P = 0.29$. (I) PWL. Ipsilateral: $F_{(2, 18)} = 2.25$, $P = 0.13$; Contralateral: $F_{(2, 18)} = 0.1$, $P = 0.9$. **(J–L)** Example heat maps (J), time spent (K), and velocity (L) in the blue light-paired chamber during preconditioning (day 1) and test sessions in ChR2 ($n = 7$) and eYFP ($n = 7$) mice. (K, $F_{(1, 12)} = 10.75$, $P = 0.007$; L, $F_{(1, 12)} = 10.75$, $P = 0.007$. **(M, N)** Immobility time in the TST and FST during unilateral blue or yellow light illumination of the RE-mPFC projection. (M) $F_{(2, 20)} = 3.07$, $P = 0.04$. (N) $F_{(2, 20)} = 3.89$, $P = 0.04$. ChR2: $n = 8$; eYFP: $n = 7$; NpHR: $n = 8$. **(O, P)** Raster plot showing sniffing episode (O) and total time spent sniffing (P, $F_{(2, 22)} = 8.88$, $P = 0.0015$) when ChR2 ($n = 8$), eYFP ($n = 8$) and NpHR ($n = 9$) mice encountered novel stimulus mice during blue or yellow light illumination of the mPFC-vlPAG projection. **(Q, R)** Effect of unilateral yellow light illumination of the RE-mPFC projection on mechanical PWT (Q, $F_{(5, 37)} = 38.05$, $P < 0.0001$) and thermal PWL (R, $F_{(5, 37)} = 25.74$, $P < 0.0001$) in SNI mice ($n = 9$ in NpHR, $n = 8$ in eYFP). **(S–U)** Representative heat maps (S), time spent (T, $F_{(1,13)} = 5.11$, $P = 0.042$), and velocity (U, $F_{(1,13)} = 0.076$, $P = 0.79$) in the yellow light-paired chamber during preconditioning (day 1) and test sessions in eYFP ($n = 7$) and NpHR ($n = 8$) SNI mice. **(V, W)** Raster plot showing sniffing episode (V) and total time spent sniffing (W, $t = 3.75$, $P = 0.002$) when eYFP ($n = 8$) and NpHR ($n = 8$) SNI mice encountered novel stimulus mice during yellow light illumination of the RE-mPFC projection. **(X, Y)** Summary of immobility time in the TST (X, $t = 4.22$, $P = 0.0009$) and FST (Y, $t = 2.52$, $P = 0.02$) in eYFP ($n = 8$) and NpHR SNI mice ($n = 8$) during unilateral yellow light illumination of the RE-mPFC projection. Scale bars: 100 µm. $*P < 0.05$, $**P < 0.01$, ns not significant. Two-way repeated measures ANOVAs with Tukey's post-hoc analysis for (D–I, K, L, Q, R, T, U). One-way ANOVAs with Tukey's post-hoc analysis for (M, N, P). Two-tailed unpaired $t$-tests for (W−Y). Data are available in S1 Data as a part of Supporting information.

Collectively, these results suggest that the RE-mPFC circuit, consisting of RE neurons, mPFC-vlPAG neurons, and mPFC GABAergic neurons, undergoes significant modification in neuropathic pain state, resulting in inhibition of mPFC-vlPAG neurons.

### The RE-mPFC projection modulates pain thresholds and depression-like behaviors in naïve and SNI mice

We subsequently investigated whether direct modulation of the RE-mPFC pathway could exert effects similar to the modulation of RE neurons. To address this, we injected AAV-CaMKII-ChR2-eYFP, AAV-CaMKII-NpHR-eYFP, or AAV-CaMKII-eYFP into the RE in mice and implanted an optical fiber in the mPFC of the right hemisphere to manipulate the RE-mPFC projection (Fig 5A–5C). In naive ChR2 mice, unilateral activation of the RE-mPFC pathway induced significant decreases in mechanical PWT and thermal PWL on the ipsilateral hind paw, with no change observed on the contralateral hind paw (Fig 5D and 5E). Conversely, inhibition of the RE-mPFC pathway in NpHR mice and yellow light illumination of the mPFC in eYFP mice did not affect mechanical or thermal thresholds on either hind paw (Fig 5F–5I). Activation of the RE-mPFC pathway induced CPA, whereas inhibition of the RE-mPFC pathway did not lead to either CPP or CPA (Fig 5J–5L). Furthermore, activation of the RE-mPFC projection significantly increased immobility time in both the TST and FST (Fig 5M and 5N) and reduced social interaction (Fig 5O and 5P). In contrast, inhibition of the RE-mPFC pathway did not change performance of naïve mice in these behavioral paradigms (Fig 5M– 5P). These findings suggest that the RE-mPFC projection modulates pain thresholds, aversion, and depression-like behaviors in naïve mice, mimicking the effect of RE neurons.

To explore whether the modification of the RE-mPFC pathway of the right hemisphere is involved in the phenotype observed in SNI (on the right side) mice, we examined the effects of NpHR-mediated optogenetic inhibition of this pathway. Consistent with optogenetic inhibition of RE neurons in SNI mice, inhibition of the RE-mPFC projection significantly increased mechanical PWT and thermal PWL on the hind paw (Fig 5Q and 5R), led to CPP (Fig 5S–5U), mitigated depression-like behaviors in social interaction test (Fig 5V and 5W), the TST, and the FST (Fig 5X and 5Y) in SNI-NpHR mice.

Similar to male SNI mice, female SNI mice exhibited elevation in pain thresholds, CPP, improvement in social interaction and depression-like behaviors when the RE-mPFC projection was inhibited with optogenetic technique (S7A–S7K Fig).

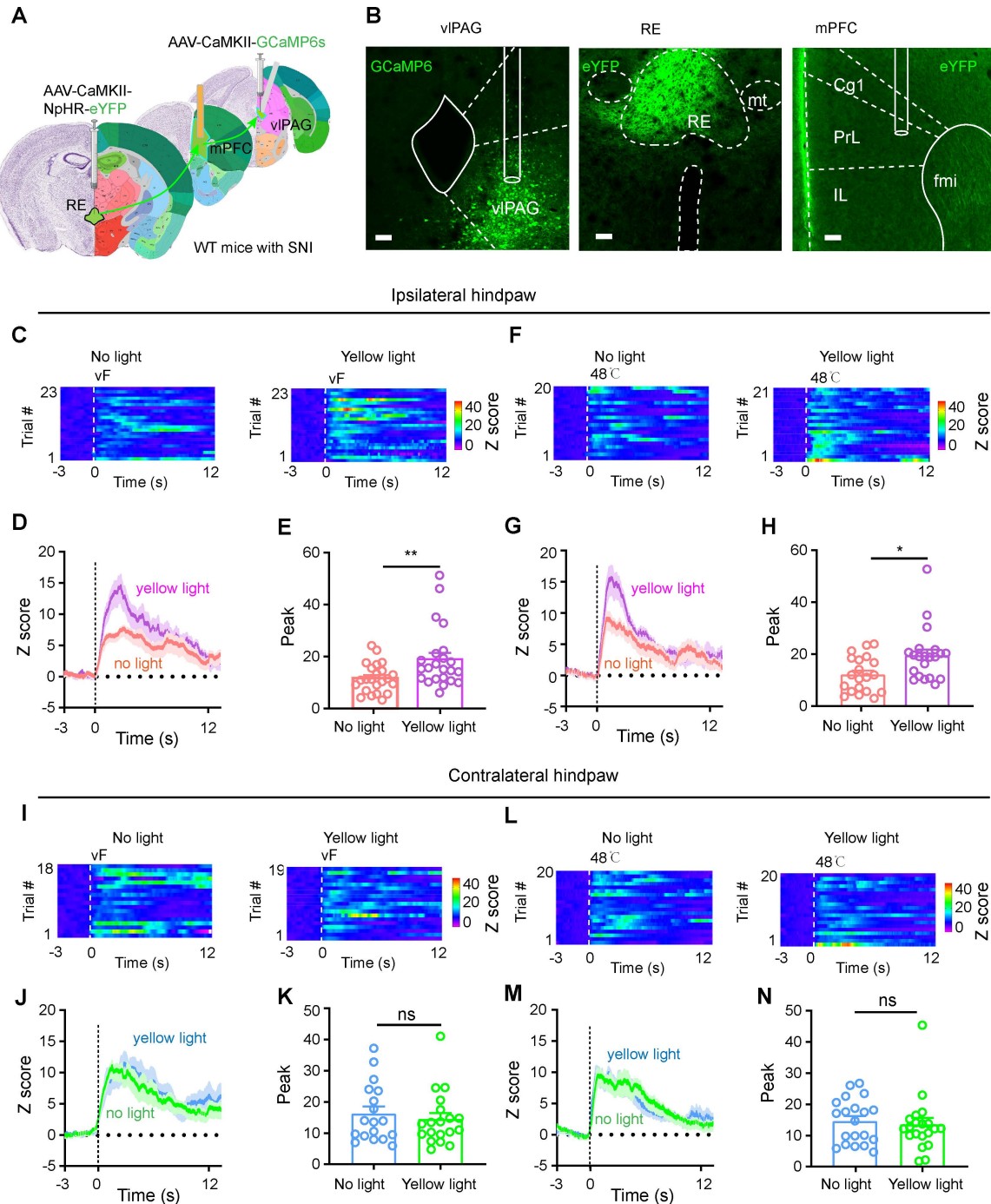

**Fig 6. Optogenetic inhibition of RE-mPFC projection enhances pain responses in vlPAG$^{Glu}$ neurons. (A)** Experimental strategy for GCaMP6 signal recording of vlPAG$^{Glu}$ neurons in SNI mice with or without optogenetic inhibition of the RE-mPFC projection. Nissl (left part of slices) and anatomical annotations (right part of slices) from the Allen Mouse Brain Atlas (https://mouse.brain-map.org) and Allen Reference Atlas-Mouse Brain (https://atlas.brain-map.org). **(B)** Example images of NpHR and GCaMP6 expression in the injection sites. **(C−N)** Heat maps (C, F, I, and L), average traces (D, G, J, and M), and summary (E, H, K, and N) of GCaMP6s signal in vlPAG$^{Glu}$ neurons of mice receiving von Frey filament (0.4 g) or thermal stimulation on ipsilateral hind paw with or without yellow light illumination of the mPFC. (E) $t = 2.72$, $P = 0.009$; (H) $t = 2.59$, $P = 0.014$; (K) $t = 0.64$, $P = 0.53$; (N) $t = 0.027$, $P = 0.79$. $n = 18−23$ trials from 5 mice in each group. Scale bars: 100 μm. Dashed lines indicate stimulus onset. *$P < 0.05$, **$P < 0.01$. Two-tailed unpaired $t$ test for (E, H, K, and N). Data are available in S1 Data as a part of Supporting information.

Collectively, these results indicate that the hyperactivity of the RE-mPFC projection is involved in chronic pain and comorbid depression in both male and female mice.

## The RE-mPFC pathway enhances nociceptive response in vlPAG Glu neurons

We further determined how inhibition of the RE-mPFC pathway affects the nociceptive responses of Glu neurons in the vlPAG, as these neurons are essential for triggering pain relief [31–33]. To achieve this, GCaMP6s was selectively expressed in vlPAG Glu neurons by injecting AAV-CaMKII-GCaMP6s into the vlPAG and AAV-CaMKII-NpHR-eYFP was injected into the RE in mice (Fig 6A). The specificity of CaMKII promoter was verified by immunostaining with a GABA antibody. We did not observe GCaMP6 expression in GABA(+) neurons in vlPAG (S8A and S8B Fig). We implanted one optical fiber into the vlPAG for fiber photometry recording of GCaMP6 signal and another into the layer 5 of the mPFC to deliver yellow light for optogenetic inhibition of the RE-mPFC projection (Fig 6A and 6B). By c-Fos-antibody staining, we confirmed that optogenetic inhibition of the RE-mPFC projection increased the number of activated CaMKII(+) neurons in the vlPAG (S8C and S8D Fig). We then performed fiber photometry recording and optogenetic modulation in these mice. We observed that stimulation with 0.4 g von Frey filament or a 48°C heating block on either ipsilateral or contralateral hind

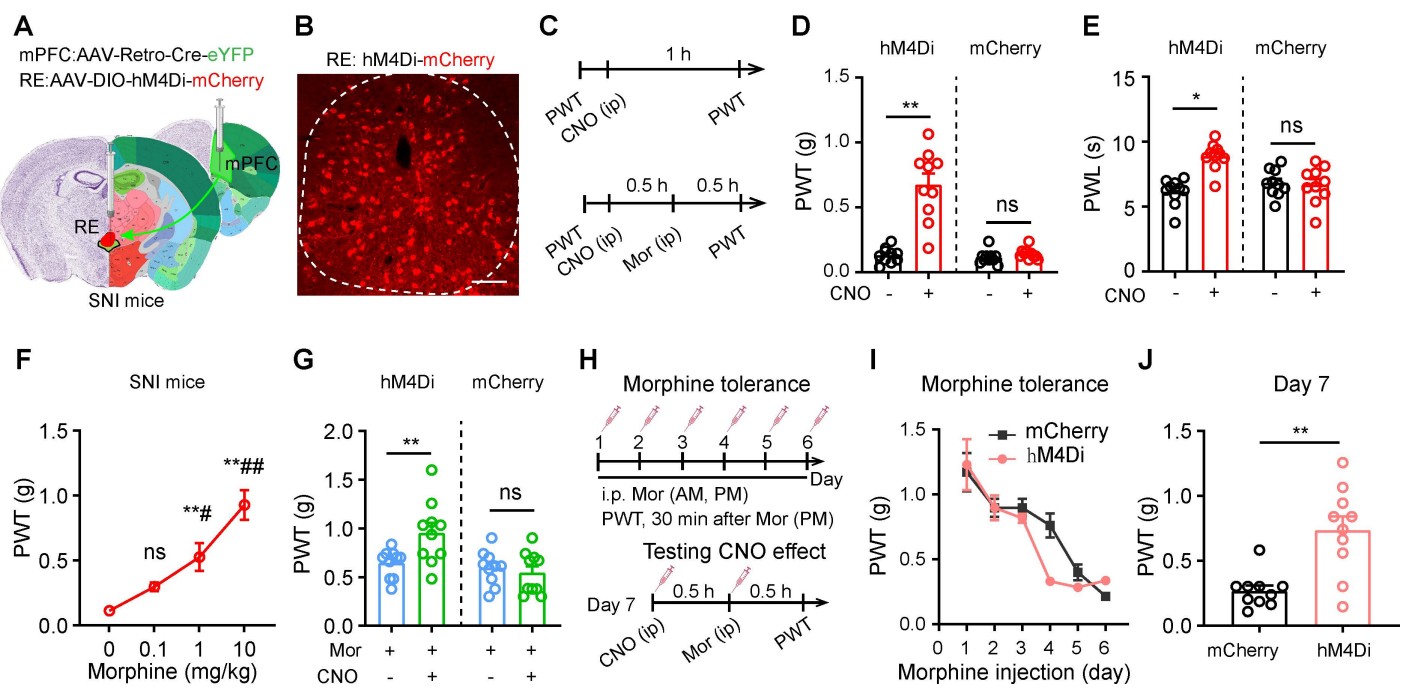

**Fig 7. Inhibition of the RE-mPFC projection elevates pain thresholds in mice subjected to acute and chronic morphine administration. (A)** Diagram of chemogenetic inhibition of RE-mPFC neurons in mice undergoing ipsilateral SNI. Nissl (left part of slices) and anatomical annotations (right part of slices) from the Allen Mouse Brain Atlas (https://mouse.brain-map.org) and Allen Reference Atlas-Mouse Brain (https://atlas.brain-map.org). **(B)** Example images of hM4Di-mCherry expression in the RE. **(C)** Timeline for examining antinociceptive effect of CNO (3 mg/kg) alone or in combination of morphine (1 mg/kg). **(D, E)** Effect of CNO on mechanical PWT (D, $F_{(5, 37)}$ = 38.05, $P<0.0001$) and thermal PWL (E, $F_{(5, 37)}$ = 25.74, $P<0.0001$) in SNI mice ($n=10$ mice in each group). **(F)** Dose–response relationship of morphine-induced analgesic effect measured with the von Frey filament test in SNI mice. $F_{(3, 20)}$ = 18.64, $P<0.0001$, $n=10$ mice in each group. **(G)** The effect of CNO combined with morphine on mechanical PWT in SNI mice ($n=10$ in each group). $F_{(5, 37)}$ = 38.05, $P<0.0001$. **(H)** Experimental protocol for panels **(I, J)**. **(I)** hM4Di (Di) and mCherry mice develop significant tolerance to 10 mg/ kg morphine twice daily for 6 days. $F_{(1, 18)}$ = 1.33, $P=0.26$, $n=10$ mice in each group. **(J)** The effect of CNO on mechanical PWT on day 7, 30 min after morphine administration in SNI mice ($n=10$ mice in each group). $t=2.72$, $P=0.009$. Scale bars: 100 μm. *$P<0.05$, **$P<0.01$, ns not significant. Two-way repeated measures ANOVAs with Tukey's post-hoc analysis for (D−G, I). Two-tailed unpaired $t$ test for (J). Data are available in S1 Data as a part of Supporting information.

paw triggered time-locked increase in GCaMP6 signal in vlPAG Glu neurons (Fig 6C–6N). Notably, optogenetic inhibition of the RE-mPFC projection significantly enhanced GCaMP6 responses to mechanical and thermal stimulation on the ipsilateral side, but not on the contralateral side (Fig 6C–6N). These results suggest that inhibition of the RE-mPFC projection amplified activation of vlPAG Glu neurons in response to pain stimuli. Since activation of vlPAG Glu neurons confers pain relief [32,33], these data may provide cellular and circuit bases for analgesic effect of RE neurons and the RE-mPFC pathway in SNI mice.

## Inhibition of the RE-mPFC pathway exerts analgesic effect in SNI mice receiving acute and repetitive exposure to morphine

Given that the RE-mPFC pathway modulates pain response in vlPAG Glu neurons and the vlPAG is a crucial site for opioid-induced analgesia [34–36], we next examined whether the RE-mPFC projection regulates pain threshold in SNI mice subjected to acute and chronic morphine exposure. To inhibit the RE-mPFC pathway, we transfected hM4Di into mPFC-projecting RE neurons by injecting AAV retro-hSyn-mCherry-Cre into the mPFC and AAV-EF1α-DIO-hM4Di-mCherry or AAV-EF1α-DIO-mCherry into the ipsilateral RE (Fig 7A and 7B). Consistent with optogenetic experiments in SNI mice (Figs 2 and 5), chemogenetic inhibition of RE-mPFC neurons with i.p. administration of clozapine-N-oxide (CNO) (3 mg/kg) reliably increased both mechanical PWT and thermal PWL on the ipsilateral hind paw in SNI-hM4Di mice, but not in SNI-mCherry mice (Fig 7C–7E). We then examined a dose-response relationship of the anti-nociceptive effect of morphine in SNI mice. As illustrated in Fig 7F, i.p. injection of morphine (0.1, 1, 10 mg/kg) caused dose-dependent increases of mechanical PWT in SNI mice, with 10 mg/kg morphine elevating PWT in SNI mice to the level similar to sham mice. Importantly, chemogenetic inhibition of RE-mPFC neurons exaggerated the elevation of mechanical PWT by submaximal morphine (1 mg/kg) in SNI mice (Fig 7G). These results indicate that inhibiting the RE-mPFC pathway enhances the analgesic effect of submaximal dose of morphine.

We then investigated the analgesic effects of RE-mPFC neurons in SNI mice undergoing morphine tolerance. After hM4Di and mCherry SNI mice (4 weeks after surgery) received morphine (10 mg/kg) intraperitoneally twice daily for 7 days (Fig 7H), morphine-induced antinociception gradually diminished (Fig 7I). On day 7, 10 mg/kg morphine provided minimal analgesia in SNI mice, however, CNO (3 mg/kg) significantly increased mechanical PWT in morphine-treated hM4Di mice but not in morphine-treated mCherry mice (Fig 7J). These results indicate that inhibition of the RE-mPFC pathway exerts analgesic effect in SNI mice even when morphine tolerance develops.

To examine whether this analgesic effect depends on endogenous opioidergic system, we tested the effect of optogenetic inhibition of the RE-mPFC projection on SNI mice with and without naloxone administration. Naloxone administration (2 mg/kg, i.p.) had no effect on either mechanical PWT or thermal PWL and did not eliminate the analgesic effect in SNI mice following inhibition of the RE-mPFC projection (S9A and S9B Fig). This result excludes the involvement of endogenous opioidergic system.

To further evaluate the analgesic effects of the inhibition of the RE-mPFC projection in SNI mice that develop morphine tolerance, we transfected NpHR into RE Glu neurons, implanted an optical fiber into the mPFC (S8A Fig), and repetitively administered morphine (10 mg/kg, twice daily) in SNI mice (Fig 7H). SNI-NpHR mice developed morphine tolerance (S9C Fig). In these mice, optogenetic inhibition of the RE-mPFC projection increased mechanical PWT (S9D Fig). These data further confirm the analgesic effect of RE-mPFC inhibition in morphine-tolerant mice.

To investigate how chronic morphine treatment modifies the RE-mPFC projection, we used the viral labeling strategy shown in S5A Fig to label mPFC GABA neurons and mPFC-vlPAG neurons, and to transfect ChR2-eYFP into RE neurons. These mice were subjected to SNI surgery, then received administration of morphine (10 mg/kg, i.p.) or saline twice daily for 7 days (S10B Fig). Paired blue light pulses separated by 50 ms evoked dual EPSCs in both mPFC-vlPAG neurons (S10C Fig) and mPFC GABA neurons (S10F Fig). The amplitudes of the first EPSCs in both vlPAG-projecting neurons and GABAergic neurons in the mPFC were much larger in morphine-tolerant SNI mice relative to those

received 7 days saline administration (S10D and S10G Fig). PPRs in mPFC-vlPAG neurons and mPFC GABA neurons in morphine-tolerant SNI mice showed a reduction trend but without reaching statistical significance, relative to saline-injected mice (S10E and S10H Fig). These results suggest that chronic morphine treatment enhances RE-mPFC synaptic transmission probably through postsynaptic mechanisms.

These results suggest that although the vlPAG is a shared nucleus for the RE-mPFC pathway and morphine to confer analgesic effect, non-opioid machinery may be recruited by the RE-mPFC pathway to confer the analgesic effect.

## Discussion

Understanding the intricate circuitry involved in emotional impacts of chronic pain is essential for developing effective treatments. While the RE-mPFC connectivity is known [22], its role in regulating chronic pain-associated behaviors remains unclear. Using fiber photometry, we linked RE neuron activity to nociception and emotional changes. Activating RE Glu neurons and their projections to the mPFC induced hypersensitivity, aversion and depression-like behaviors in naïve mice. Conversely, inhibiting this pathway alleviated these symptoms in mice with chronic pain. Examination of mPFC-vlPAG neurons revealed that chronic neuropathic pain enhanced the RE-mPFC projection favoring mPFC GABAergic circuit over mPFC-vlPAG neurons, leading to hypoexcitability in mPFC-vlPAG neurons. Inhibiting the RE-mPFC Glu pathway increased the activity in vlPAG Glu neurons, and ameliorated hyperalgesia and depression-like behaviors in neuropathic pain mice. Additionally, inhibition of the RE-mPFC pathway conferred analgesia in SNI mice after acute and chronic morphine treatment. These findings shed light on how the RE-mPFC pathway contributes to chronic pain-related behaviors.

The RE is anatomically connected to various cortical and subcortical brain regions involved in emotion and pain signal processing [37]. Consistent with this notion, our in vivo fiber photometry recording data show that the activity of RE neurons is sensitive to and is increased by noxious stimulation, environmental settings associated with anxiety (open arms in the EPM), social interaction and aversion (air puff). When activated, RE neurons triggered hyperalgesia, aversion- and depression-like behaviors. These observations collectively suggest the multifaceted roles of RE neurons in pain sensation and associated emotional processing. In the future, it is worthwhile to examine how the multifaceted signals are relayed to the RE.

Previous studies have highlighted the mPFC-RE projection and its role in memory and emotional processing [22,24], however, the role of the RE-mPFC projection has received less attention. In this study, we investigated how RE Glu neurons modulate mPFC neurons and related behaviors. We demonstrated that stimulation of the RE-mPFC Glu projection elicited hyperalgesia and depression-like behaviors similar to direct stimulation of RE Glu neurons. These results suggest a crucial role that the mPFC plays in mediating the function of the RE in these domains. Overall, the reciprocal communication between the RE and mPFC likely contributes to the fine tuning of both pain signaling and emotional processing.

Studying the neural circuit involved in chronic pain helps identify shared brain regions associated with pain behavior and psychiatric disorders. Malfunction in mPFC circuitry is importantly involved in the pathophysiology of chronic pain [4,6,8,10,13,38]. Inhibition of mPFC input nuclei, including the mediodorsal thalamus, ventral tegmental area (VTA), and other relevant regions, ameliorates neuropathic pain [8,19,20,38]. We observed that in neuropathic pain mice, RE Glu neurons displayed hyperactivity and the RE-mPFC projection was enhanced; inhibition of the RE-mPFC pathway to reverse these changes effectively alleviate hyperalgesia and depression-like behaviors. Our data addressed pathophysiological alterations in mPFC circuit and provide an explanation for amelioration of neuropathic pain by inhibition of Glu inputs to the mPFC.

In SNI mice, despite the enhancement of the RE-mPFC projection, optogenetic stimulation of the RE-mPFC Glu projection resulted in reduced firing of mPFC-vlPAG neurons, which also exhibited hypoexcitability. We postulate that mPFC GABAergic interneurons may be major contributors for this phenomenon. Although we did not manipulate mPFC GABAergic interneurons in SNI mice, we observed that SNI enhanced synaptic inputs from the RE to mPFC GABAergic

neurons. Additionally, optogenetic stimulation of the RE-mPFC projection evoked larger IPSCs in mPFC-vlPAG neurons in SNI mice than in sham mice, resulting in a larger I/E ratio. This result may be explained by the enhancement of inhibitory polysynaptic inputs to mPFC projection neurons mediated by mPFC GABAergic interneurons. This idea is consistent with the notion that the maintenance of baseline pain perception and emotion is associated with persistent activity of mPFC Glu neurons [16,39–41]. Moreover, our findings extend the framework proposed by Huang and colleagues [8] to a new projection (the RE-mPFC pathway) and suggest that feedforward inhibition in the mPFC serves as a common mechanism for gating pyramidal neuron output to regulate distinct behavioral paradigms, such as chronic pain and emotional responses. Both the RE projections investigated in our study and the BLA projections in the study conducted by Huang and colleagues [8] converge on a key mechanism: increased excitatory inputs to the mPFC does not directly activate pyramidal neurons but instead amplify inhibitory control, thereby suppressing mPFC output, such as its regulation of the vlPAG in pain processing.

The activation of RE neurons induces hyperalgesia in both hind paws, while activation of unilateral RE-mPFC projection causes hyperalgesia on the ipsilateral hind paw, aligning with the RE's bilateral communication [21,24] and the known antinociceptive effects of mPFC projection neurons likely through connections with the PAG. Although inhibiting RE neurons or their inputs to the mPFC in control mice shows no discernible effect on pain-related behaviors in naïve mice, their activation leads to pain- and depression-like behaviors. These results suggest that these circuit components may not be necessary for maintaining normal pain thresholds and emotional states, but activation of them may significantly override the existing circuitry and change these behaviors.

We demonstrated that inhibition of the RE-mPFC pathway enhanced pain responses of vlPAG Glu neurons to stimulation of the ipsilateral hind paw in neuropathic pain mice. This finding aligns with the known lateralization of descending pain modulatory pathways, where the vlPAG primarily operates in an ipsilateral manner [8,14]. In the context of neuropathic pain, this ipsilateral-specific enhancement of vlPAG Glu responses upon RE-mPFC inhibition could reflect circuit-level adaptations or sensitizations that reinforce pain processing on the injured side. These results highlight the role of lateralized dynamics within a tertiary neural circuit (RE$^{Glu}$→mPFC$^{Glu}$/(mPFC$^{GABA}$-mPFC$^{Glu}$) → vlPAG$^{Glu}$ neurons) in pain modulation.

In this study, we extended the potential application of modulation of the RE-mPFC pathway in the treatment of neuropathic pain. Opioid analgesics have been commonly prescribed in clinical settings. Rapid development of tolerance and life-threatening side-effects of these drugs at higher doses limited their applications [42,43]. Non-opioid drugs either strengthen or exert comparable analgesic effect of opioid analgesics have been sought to solve the challenge in the treatment of chronic pain. We provide evidence for the notion that inhibition of the RE-mPFC pathway may be a potential therapeutic strategy, because it enhanced analgesic effect of submaximal dose of morphine in SNI mice and elevated pain threshold even when SNI mice developed morphine tolerance.

In summary, our findings suggest that the RE-mPFC pathway is involved in the pathophysiology of neuropathic pain comorbid with depression. The enhanced RE-mPFC pathway preferentially activates mPFC GABAergic circuit over mPFC-vlPAG neurons, leading to reduced neuronal firing in the latter. Inhibition of this pathway reverses hypoactivity in both mPFC-vlPAG neurons and vlPAG neurons, resulting in analgesic and antidepressant effects. Additionally, inhibition of the RE-mPFC Glu pathway may provide pain relief through a non-opioid mechanism. Therefore, the RE-mPFC Glu pathway represents a potential non-opioid therapeutic target for treating neuropathic pain and its comorbid emotional disorders.

## Materials and methods

### Ethics statement

The care and use of animals and the experimental protocols (No. 202207S123) used in this study were approved by the Institutional Animal Care and Use Committee and the Office of Laboratory Animal Resources of Xuzhou Medical University under the Regulations for the Administration of Affairs Concerning Experimental Animals (1988) in China.

## Animals

Wild-type C57BL/6 (WT) mice, and CaMKII-Cre and Vgat-Cre mice with C57/BL6 background were bred in-house in the animal facility of Xuzhou Medical University. Mice were group-housed (no more than 5 per cage) in a temperature- and humidity-controlled housing facility on a 12-h light/dark cycle with ad libitum access to food and water. Male mice at least eight weeks old were used in the experiments. Efforts were made to minimize animal suffering and to reduce the number of animals used.

## Viral vectors

AAV-CaMKII-GCaMP6s, AAV-CaMKII-eYFP, AAV-CaMKII-ChR2-eYFP, AAV-CaMKII-NpHR3.0-eYFP, AAV retro-hSyn-Cre-eYFP, AAV retro-hSyn-Cre-mCherry, AAV-EF1α-DIO-hM4Di-mCherry, AAV-Ef1α-DIO-TVA-GFP, AAV-Ef1α-DIO-RVG, RV-EnvA-ΔG-DsRed were purchased from Brain VTA (Wuhan, China). The viral titers are $2-5 \times 10^{12}$ (viral genomes per ml) for AAV2/9 and $1-5 \times 10^{8}$ (viral genomes per ml) for rabies virus (RV).

## Stereotaxic surgeries and viral injection

Mice were deeply anesthetized with isoflurane (3% for induction and 1.5% for maintenance), placed on a heating pad, and secured on a stereotaxic apparatus (RWD Life Science Co., Ltd, Shenzhen, China). A small midline dorsal incision was made to expose the skull and then small holes were drilled in the skull above brain regions of interest. Unilateral viral injections (120 nl of virus per site at 50 nl/min) were performed with a microinjection pump (KD Scientific, Holliston, MA, USA) using the following coordinates: (relative to the Bregma): RE (AP, −1.30 mm; ML, 0.63 mm; DV, 4.58 mm), mPFC (AP, +1.70 mm; ML, 0.3 mm; DV, 2.50 mm), and vlPAG (AP, −4.70 mm; ML, 0.5 mm; DV, 3.30 mm). Optical fiber implants (200 μm in diameter, NA 0.37) (Inper, Hangzhou, China) were placed 200 μm above (for optogenetic manipulation) or in (for fiber photometry recording) the injection site and were fixed to the skull with dental cement. Mice with virus injections and optical fiber implants were allowed to recover for at least 3 weeks before electrophysiological recordings and morphological assays. Viral expression and the position of fiber implants in each mouse were confirmed histologically after the termination of the experiments. We only included mice with viral expression confined to the RE, mPFC, or vlPAG and optical fibers in the right places for either optogenetic modulation or fiber photometry recording.

Meloxicam (4 mg/kg) (Aladdin Biochemical Technology, Shanghai, China) was administered subcutaneously once per day for 3 days for postoperative pain relief.

## Fiber photometry

A fiber photometry instrument (ThinkerTech, Nanjing, China) [28,44–46] was used to monitor GCaMP6 and eYFP signals in RE and vlPAG neurons. We adjusted the instrument by setting the excitation light to 50 μW and the gain to a level that gave a background signal of 3 units. The input optical cable was then connected to the optical implant in the mouse brain. To evaluate responses of RE and vlPAG neurons to mechanical, thermal, and emotional stimulation, we designated 3 s GCaMP6 and eYFP signal prior to the stimulation as the baseline value ($F_0$) and calculated the mean and standard deviation (SD) of $F_0$. The GCaMP6 and eYFP signals ($F$) were transformed into z-scores using the formula $((F - \text{Mean}[F_0])/\text{SD}[F_0])$. The peak of the z-score plot was measured to quantify the neuronal responses to sensory and emotional stimulation.

## Optogenetic manipulation

For ChR2-mediated optogenetic stimulation, 473 nm laser pulses (5 ms, 20 Hz, 4 mW) were delivered. For NpHR-mediated optogenetic inhibition, a 589-nm laser was kept on continuously for 1–2 min (constant, 3 mW). All optogenetic manipulations were performed unilaterally in the right hemisphere. Therefore, the contralateral and ipsilateral sides refer to the left and the right sides, respectively.

## Chemogenetic manipulation

Three weeks after virus injection and SNI, mice were injected intraperitoneally with clozapine-N-oxide (CNO, 3 mg/kg) 60 min before nociceptive threshold measurement. Baselines for nociceptive thresholds were measured 1 h before CNO administration.

## von Frey filament test

Individual mice were acclimatized for at least 1 h in a test compartment on a wide gauge wire mesh supported by an elevated platform. The von Frey filaments with fiber forces between 0.01 and 2 g were used to measure mechanical PWT of both hind paws. The 50% threshold was determined with the up-down method [47,48].

## Thermal nociception threshold

Individual mice were acclimatized for at least 1 h in a test compartment on a glass surface. The Hargreaves test was performed to evaluate the thermal PWL. We applied the radiant heat source (Boerni, Tianjin, China) to the hind paw and measured the latency to evoke a withdrawal. Three replicates were acquired, and values were averaged per hind paw per mouse [48,49].

## Spared nerve injury (SNI)

Chronic neuropathic pain model was established with SNI of the sciatic nerve according to a previously reported protocol [48,50]. Mice were deeply anesthetized using isoflurane (3% for induction, 1.5% for maintenance) (RWD Life Science Co., Ltd, Shenzhen, China). The fur in the surgical area, extending from the knee to the hip, was shaved, and the skin was sterilized with 75% alcohol. A longitudinal incision was made in the shaved region, allowing for blunt dissection of the biceps femoris muscle to expose the sciatic nerve and its branches (sural, common peroneal, and tibial nerves). The surgery was performed on the same side as the viral injection (right side). Two nylon sutures 3 mm apart were tightly ligated around the common peroneal and tibial nerves, and the nerves between the sutures were subsequently cut and removed. The mice were then allowed to recover on a heating pad. To serve as control, mice that did not receive nerve ligation and severing were used as sham controls. Pain thresholds were assessed using von Frey filaments and a heating beam targeting the skin area innervated by the sural nerve.

## Conditioned place preference (CPP) test

The CPP was performed in a custom-made two-chamber box (length × width × height: $40 \times 20 \times 30\,cm^3$): the right chamber had vertical black-and-white stripes on the walls and a smooth floor, whereas the left chamber had horizontal black-and-white stripes on the walls and a mesh floor. The CPP test was performed according to a previous study with some modifications [46].

 Day 1 was the preconditioning test (pre-test) day. Mice were given free access to the two chambers and the time the mice spent in each chamber was recorded. On day 2–4, the mice were restricted to one chamber for 20 min and received optogenetic modulation in the morning. At least 4 h later, the mice were restricted to the other chamber for 20 min without optogenetic modulation in the afternoon. On day 5 (around 24 h after the last conditioning), the mice were allowed to freely explore the two chambers for 20 min and the time spent in each chamber was recorded. On the precondition and test days, the mouse behavior was recorded with a video-camera and the data were analyzed online or offline with an EthoVision XT video tracking software (Noldus Information Technology, Wageningen, Netherland) [51,52]. We calculated the time spent in the light-paired side on the precondition and test days. Mice were not used if they spent more than 75% of the total time in one chamber on the precondition day.

## Measurement of depression-like behaviors

Depression-like behaviors were assessed by TST, FST, and social interaction test [28,53], and these behavioral assessments in SNI mice were conducted 4−5 weeks after SNI surgery.

Tail suspension test (TST).A mouse was suspended by taping its tail onto a horizontal bar 50 cm above the floor. The mouse was allowed to hang undisturbed for 6 min and its behavior was video-recorded. The total duration that the mouse remained immobile in the last 5 min was used to evaluate depression-like behavior.

Forced swim test (FST).Mice were placed in a one-liter glass beaker filled with 26°C water. The movement of the mice was recorded for 5 min using a camera placed beside the beaker, and the immobility time was measured.

Social interaction testThe social interaction test was used to evaluate the social motivation of mice. A novel conspecific mouse was introduced into the home cage of the test mouse for 5 min, during which the time spent in social interaction, including sniffing the body parts and making contacts, was recorded.

## Brain slice electrophysiology

Brain slice electrophysiological recording was conducted with minor modifications according to previously reported methods [28,46,52]. Coronal slices (250–300 μm thick) containing the RE or mPFC were prepared using a vibratome (Leica VT-1200S, Nussloch, Germany) in an ice-cold cutting solution saturated with 95% $O_2$/ 5% $CO_2$ (carbogen), containing (in mM) 85 NaCl, 75 sucrose, 2.5 KCl, 1.25 $NaH_2PO_4$, 4.0 $MgCl_2$, 0.5 $CaCl_2$, 24 $NaHCO_3$, and 25 glucose. The brain slices were transferred into carbogenated cutting solution at 32°C and allowed to recover for 60 min, and then placed in normal carbogenated ACSF containing (mM) 125 NaCl, 2.5 KCl, 1.2 $NaH_2PO_4$, 1.2 $MgCl_2$, 2.4 $CaCl_2$, 26 $NaHCO_3$, and 11 glucose at 26°C for at least 30 min prior to use.

Neurons in brain slices were visualized under an upright microscope (FN-1, Nikon, Tokyo, Japan), equipped with a CCD-camera (Flash 4.0 LTE, Hamamatsu, Hamamatsu city, Japan). Whole-cell patch-clamp recordings were obtained using a patch-clamp setup composed of a dual-channel MultiClamp 700B amplifier, a Digidata 1550B analog-to-digital converter, and pClamp 10.7 software (Molecular Devices, San Jose, CA, USA). The patch electrodes had a resistance of 4−6 MΩ when filled with a low-chloride intrapipette solution containing (in mM) 135 K gluconate, 0.2 EGTA, 0.5 $CaCl_2$, 10 HEPES, 2 Mg-ATP, and 0.1 GTP, pH: 7.2; osmolarity: 290−300 mOsm. Neurons with a holding current larger than −50 pA and a resting membrane potential more depolarized than −40 mV were excluded from the analysis. All recordings were performed at 32 ± 1°C.

For light-evoked responses, blue light (460 nm, 2 mW) or yellow light (560 nm, 2 mW) was delivered through an optical fiber (200 μm, NA 0.37) connected to a PlexBright LED light source (Plexon, Hong Kong, China). The light-evoked EPSCs and IPScs were recorded at −70 mV and 0 mV, respectively. To confirm whether Glu and GABAergic connections were involved, CNQX (20 μM) or BIC (10 μM) was bath-applied, respectively. Firing in response to current injections (1 s, 20−200 pA steps with an increment of 20 pA and a 30 s inter-sweep interval) were recorded in the current-clamp mode.

## Immunohistochemistry

Mice were sacrificed in a $CO_2$ chamber and then subjected to cardiac perfusion with phosphate-buffered saline (PBS), followed by 4% paraformaldehyde (PFA) in PBS. Mouse brains were removed and post-fixed in 4% PFA overnight at 4°C and then immersed in 30% sucrose in PBS until sank for cryoprotection. Using a Leica CM1950 cryostat (Nussloch, Germany), 30 μm tissue sections were prepared and mounted onto glass slides. For immunostaining, brain sections on glass sides were incubated in a blocking buffer containing 5% donkey serum and 0.1% Triton X-100 for 90 min at room temperature. Then the sections were incubated with primary antibody diluted in blocking buffer for 24 h at 4°C (rabbit anti-c-Fos IgG, 1:2000, Cell Signaling Technology; Mouse anti-CaMKII IgG, 1:1000, Cell Signaling Technology). After washing three times (10 min each) in PBS, the sections were incubated with secondary antibodies (Alexa 488- or Alexa 647-conjugated

donkey anti-rabbit IgG, Alexa 555-conjugated donkey anti-mouse IgG) for 90 min at room temperature. The sections were washed three times (10 min each) in PBS, dried in the dark, and then cover-slipped in mounting medium (Meilunbio, Dalian, China).

The sections were imaged with a confocal microscope (Fv-1000, Olympus), and the images were processed with ImageJ (NIH) [54].

## Morphine tolerance

We performed morphine tolerance in SNI mice with RE-mPFC projecting neurons expressing mCherry or hM4Di by administering 10 mg/kg morphine (i.p.) twice per day, separated by approximately 8 h. We measured mechanical PWT before the first injection and 1 h after morphine injection every day to verify the development of morphine tolerance. On day 7, a combined injection of 10 mg/kg morphine and 3 mg/kg CNO was administered in mCherry and hM4Di mice, mechanical PWT was measured 1 h after administration.

## Chemicals

4-AP, bicuculline methochloride (Bic), CNQX, naloxone hydrochloride, and TTX were purchased from Tocris. CNO was purchased from MedChemExpress and administrated intra peritoneally (i.p.) at a dose of 3 mg/kg. Morphine was purchased and used as a controlled substance under restricted supervision. It was administrated i.p. across a dose ranging from 0.1 to 10 mg/kg.

## Statistical analysis

GraphPad Prism 8.0 was used for statistical analysis. Clampfit 10.7 (Molecular Devices) was used for analysis of electrophysiological and GCaMP6 data. Figures were prepared with Adobe Illustrator 2020. Data analysis and presentation complied with a previously published guideline [55]. All summarized data are expressed as mean ± S.E.M. Two-tailed paired or unpaired $t$-tests were used for comparison of a parameter between two groups if data were normally distributed. One-way ANOVA followed by Tukey's post-hoc analysis was used for multiple comparisons. If the equal-variance assumptions were not valid, statistical significance was evaluated with the Mann–Whitney test or ANOVA rank tests. The mean and S.E.M, $n$ (the number of animals), statistical test, and $t$, F, and P values are reported in the figure legends. A value of $P < 0.05$ was considered statistically significant. The minimal number of mice used in each experiment was calculated in a priori power analysis (StatMate 2.0) and the power of each experiment was set to 0.8. The sample sizes in experiments are larger than the minimal numbers.

## Supporting information

**S1 Fig. Pain stimulation induces increased activity in RE glutamatergic neurons in naïve mice. (A, B)** Representative images and summary showing that injection of AAV-CaMKII-ChR2-eYFP into the RE transfected eYFP into the RE neurons, most of which were immunofluorescently stained with the CaMKII-antibody (red). $n = 4$ mice. **(C, D)** Representative images showing the labeling of RE neurons with AAV-CaMKII-eYFP and c-Fos-antibody-staining (red) in mice whose hind paws received (C, vF(+)) or did not receive (D, vF(−)) stimulation of the 2 g von Frey filament. Arrows indicate overlapping of c-Fos with eYFP. **(E)** Quantification of percent of eYFP-expressing RE neurons stained with c-Fos-antibody. $t = 2.62$, $P = 0.02$, $n = 8$ mice. Two-tailed unpaired $t$ test. Data are available in S1 Data as a part of Supporting information. vF, von Frey filament. Scale bar: 100 μm. (TIFF)

**S2 Fig. SNI mice develop hyperalgesia and depression-like behavior. (A, B)** Time courses of mechanical withdrawal threshold (PWT) (left panel) and thermal (right panel) withdrawal latency (PWL) after SNI and sham surgery. (A) $F_{(5, 70)} =$

20.8, $P < 0.0001$. (B) $F_{(5, 70)} = 65.16$, $P < 0.0001$. $n = 8$ in each group. **(C, D)** Immobility time in the tail suspension test (TST) (C, $t = 2.79$, $P = 0.01$, $n = 8$ in each group) and the forced swim test (FST) (D, $t = 4.5$, $P = 0.0005$, $n = 8$ in each group). **(E, F)** Total time spent sniffing the same stimulus mice by sham and SNI mice over 4 trials with 1 h intervals between trials (E, $F_{(1, 18)} = 13.26$, $P = 0.002$). (F) Total time that sham and SNI mice spent sniffing the novel stimulus mice. $t = 4.92$, $P = 0.0002$. *$P < 0.05$, **$P < 0.01$, ns not significant. Two-tailed unpaired $t$ test for (C, D, F); Two-way repeated measures ANOVAs with Tukey's post-hoc analysis for (A, B, E). Data are available in S1 Data as a part of Supporting information. (TIFF)

**S3 Fig. Effects of optogenetic modulation of RE neurons on pain thresholds. (A, B)** Effect of blue light illumination of RE neurons on mechanical PWT (A, $F_{(2, 27)} = 2.81$, $P = 0.08$) and thermal PWL (B, $F_{(2, 27)} = 0.04$, $P = 0.96$) in ChR2 mice subjected to SNI surgery ($n = 10$ SNI mice). **(C–F)** Effect of optogenetic inhibition of RE neurons on mechanical paw withdrawal threshold (PWT) (C, $F_{(2, 24)} = 0.19$, $P = 0.83$; D, $F_{(2, 24)} = 0.17$, $P = 0.85$; $n = 9$ each group) and thermal paw withdrawal latency (PWL) (E, $F_{(2, 24)} = 0.17$, $P = 0.85$; F, $F_{(2, 24)} = 0.23$, $P = 0.79$) on either hind paw of naïve NpHR mice ($n = 9$). **(G)** Schematic diagram and representative image of viral expression in the RE for optogenetic inhibition of RE neurons in SNI female mice. Left panel: Nissl (left part of slices) and anatomical annotations (right part of slices) from the Allen Mouse Brain Atlas (mouse.brain-map.org) and Allen Reference Atlas-Mouse Brain (atlas.brain-map.org). **(H, I)** Effect of NpHR-mediated inhibition of RE neurons on mechanical PWT ($F_{(2, 21)} = 25.73$, $P < 0.0001$) and thermal PWL ($F_{(2, 21)} = 29.62$, $P < 0.0001$) in SNI mice ($n = 8$ mice). **(J, K)** Example heat maps (J) and quantification of time spent (K, Time, $F_{(1, 14)} = 14.79$, $P = 0.0018$) in the yellow-light-paired chamber during the preconditioning (Pre) and test sessions for NpHR mice ($n = 8$) and eYFP mice ($n = 8$). **(L)** Velocity of NpHR mice and eYFP mice in light paired-chamber during the test session ($t = 1.97$, $P = 0.069$, $n = 8$). **(M, N)** Raster plot showing sniffing episode (M) and total time spent sniffing (N, $t = 3.02$, $P = 0.0092$) in NpHR ($n = 8$) and eYFP ($n = 8$) mice tested during yellow light illumination of the RE. **(O, P)** Immobility time in the FST (O, $t = 2.78$, $P = 0.015$) and TST (P, $t = 3.80$, $P = 0.002$) in NpHR ($n = 8$) and eYFP ($n = 8$) mice during yellow light illumination of RE neurons. *$P < 0.05$; **$P < 0.01$; ns not significant. One-way repeated measures ANOVAs for (A−F, H, I). Two-way ANOVA with Tukey's post-hoc analysis for (K). Two-tailed unpaired $t$-tests for (L, N−P). Data are available in S1 Data as a part of Supporting information. (TIFF)

**S4 Fig. RE glutamatergic neurons project to mPFC neurons. (A)** Schematic diagram of the neuronal tracing strategy for probing upstream nuclei of mPFC glutamatergic (Glu) neurons. Nissl (left) and anatomical annotations (right) from the Allen Mouse Brain Atlas (https://mouse.brain-map.org) and Allen Reference Atlas-Mouse Brain (https://atlas.brain-map.org). **(B)** The injection site and viral expression (right) in the mPFC (Arrows indicate starter cells). **(C−E)** Representative images and quantification of upstream nuclei of mPFC Glu neurons, including the RE and other brain regions. $n = 3$ mice. **(F)** Schematic diagram of the virus tracing strategy for probing the upstream nuclei of mPFC GABAergic neurons. Nissl (left) and anatomical annotations (right) from the Allen Mouse Brain Atlas (https://mouse.brain-map.org) and Allen Reference Atlas-Mouse Brain (https://atlas.brain-map.org). **(G)** The injected site and viral expression in the mPFC (Arrows indicate starter cells). **(H−J)** Representative images and quantification of upstream nuclei of mPFC GABAergic neurons, including the RE and other brain regions. $n = 3$ mice. **(K, L)** Schematic diagram for virus injection (K) and example image (L) showing that AAV-CaMKII-ChR2-eYFP was injected into the RE. **(M, N)** Example images (M) and quantification (N) of ChR2-eYFP-labeled fibers in the layers I-V of the mPFC ($n = 5$ mice). (K) Nissl (left) and anatomical annotations (right) from the Allen Mouse Brain Atlas (mouse.brain-map.org) and Allen Reference Atlas-Mouse Brain (atlas.brain-map.org). (N) The axonal terminals were quantified by pixels of eYFP. Scale bars: 100 μm. Data are available in S1 Data as a part of Supporting information. ACC, anterior cingulate cortex; BLA, basolateral amygdala; fr, fasciculus retroflexus; Hip, hippocampus; ml, medial lemniscus; mPFC, medial prefrontal cortex; RE, nucleus reuniens of the thalamus; VTA, ventral tegmental area. (TIFF)

**S5 Fig. The RE-mPFC projection innervates mPFC GABAergic interneurons. (A)** Diagram shows virus injection strategy for labeling mPFC GABA neurons and mPFC-vlPAG projecting neurons and recording their response to opto-genetic activation of RE neurons. Nissl (left part of slices) and anatomical annotations (right part of slices) from the Allen Mouse Brain Atlas (https://mouse.brain-map.org) and Allen Reference Atlas-Mouse Brain (https://atlas.brain-map.org). **(B)** Representative images of labeled neurons in the mPFC. **(C)** Representative traces of blue light-evoked EPSC (recorded at −70 mV) in eYFP-labeled GABA neurons before and after application of TTX (1 μM) and 4-AP (100 μM). **(D)** Quantification of amplitude of light-evoked EPSCs on labeled GABA neurons before and during application of TTX and 4-AP. $t = 1.13$, $P = 0.30$, $n = 7$ cells from 4 mice. **(E, F)** Representative traces and summary of amplitudes of eEPSCs recorded from labeled GABA neurons (−70 mV) receiving RE glutamatergic projections in SNI and sham mice. $t = 3.05$, $P = 0.005$, $n = 13 − 18$ cells from 5 mice in each group. **(G, H)** Representative traces and summaries of changes in number of firing evoked by 1 s blue light (BL) in mPFC-vlPAG neurons in sham and SNI mice. $t = 1$, $P = 0.36$, $n = 7$ cells each group. **(I, J)** Example traces and summary of frequencies of firing evoked by depolarizing current injection in mPFC GABA neurons from sham and SNI mice in the presence of 20 μM CNQX and 10 μM bicuculline. $F_{(1, 10)} = 0.25$, $P = 0.63$, $n = 8$ cells from 5 mice in each group. Scale bars: 100 μm. BL, blue light. *$P < 0.05$, **$P < 0.01$, ns not significant. Two-tailed paired $t$-tests for (D). Two-tailed unpaired $t$ test for (F, H). Two-way repeated measures ANOVA with Tukey's post-hoc analysis for (J). Data are available in S1 Data as a part of Supporting information. (TIFF)

**S6 Fig. mPFC- and hippocampus-projecting RE neurons are largely separated. (A)** Schematic diagram for labeling of upstream nuclei of the ventral hippocampus (VHip) and medial prefrontal cortex (mPFC). Nissl (left part of slices) and anatomical annotations (right part of slices) from the Allen Mouse Brain Atlas (https://mouse.brain-map.org) and Allen Reference Atlas-Mouse Brain (https://atlas.brain-map.org). **(B)** Representative coronal sections showing retrograde virus injection into the VHip (AAV retro-hSyn-mCherry) and mPFC (AAV-retro-hSyn-eYFP). **(C, D)** Example images (C) and summary (D) of RE-VHip (red) and RE-mPFC (green) neurons ($n = 4$ mice). Data are available in S1 Data as a part of Supporting information. Scale bars: 100 μm. (TIFF)

**S7 Fig. Inhibition of the RE-mPFC projection improves hyperalgesia and emotions in female SNI mice. (A)** Schematic diagram and representative image of viral expression in mPFC for optogenetic inhibition of RE-mPFC projections in SNI female mice. Nissl (left part of slices) and anatomical annotations (right part of slices) from the Allen Mouse Brain Atlas (https://mouse.brain-map.org) and Allen Reference Atlas-Mouse Brain (https://atlas.brain-map.org). **(B, C)** Effect of NpHR-mediated inhibition of RE-mPFC projections on mechanical PWT ($F_{(2, 21)} = 38.93$, $P < 0.0001$) and thermal PWL ($F_{(2, 21)} = 40.05$, $P < 0.0001$) in SNI mice ($n = 8$ mice). **(D, E)** Example heat maps (D) and quantification of time spent (E, Time, $F_{(1, 14)} = 4.15$, $P = 0.061$) in the yellow-light-paired chamber during the preconditioning (Pre) and test sessions for NpHR mice ($n = 8$) and eYFP mice ($n = 8$). **(F)** Velocity of NpHR mice and eYFP mice in light-paired chamber during the test session ($t = 0.73$, $P = 0.48$, $n = 8$). **(G–I)** Raster plot showing sniffing episode (G, H) and total time spent sniffing (I, $t = 4.35$, $P = 0.0007$) in NpHR ($n = 8$) and eYFP ($n = 8$) mice tested during yellow light illumination of the RE-mPFC projection. **(J, K)** Immobility time in the FST (J, $t = 2.77$, $P = 0.014$) and TST (K, $t = 4.46$, $P = 0.0005$) in NpHR ($n = 8$) and eYFP ($n = 8$) mice during yellow light illumination of the RE-mPFC projection. Scale bars: 100 μm. *$P < 0.05$, **$P < 0.01$, ns not significant. One-way repeated measures ANOVAs with Tukey's post-hoc analysis for (B, C). Two-way ANOVA with Tukey's post-hoc analysis for (E). Two-tailed unpaired $t$-tests for (F, I−K). Data are available in S1 Data as a part of Supporting information. (TIFF)

**S8 Fig. Optogenetic inhibition of the RE-mPFC projection enhances activity of vlPAG$^{Glu}$ neurons. (A)** Diagram of experimental design for examining c-Fos expression in the vlPAG with optogenetic inhibition of the RE-mPFC projection. Nissl (left part of slices) and anatomical annotations (right part of slices) from the Allen Mouse Brain Atlas (https://mouse.brain-map.org)

and Allen Reference Atlas-Mouse Brain (https://atlas.brain-map.org). **(B)** Representative images of an area in the vlPAG showing CaMKII-GCaMP6 expression (green) and GABA antibody staining (blue). **(C, D)** Representative images and quantification of activated Glu neurons (CaMKII-and c-Fos-reactive neurons) in SNI mice with or without yellow light stimulation of RE axonal terminals in the mPFC. $t = 5.38$, $P = 0.006$. Five PAG-containing sections from each mouse were analyzed and the average of percent of CaMKII-positive neurons labeled with c-Fos-antibody was calculated. Data from 3 mice in each group were included. Two-tailed unpaired $t$ test. **$P < 0.01$. Scale bars: 100 μm for (B), 50 μm for (C). Data are available in S1 Data as a part of Supporting information.
(TIFF)

**S9 Fig. Naloxone does not block analgesic effect by optogenetic inhibition of the RE-mPFC projection. (A, B)** The effect of naloxone on mechanical PWT and PWT before and during yellow light (YL) stimulation of the RE-mPFC projection in SNI mice. $F_{(1, 14)} = 36.52$, $P < 0.0001$ for (A); $F_{(1, 14)} = 14.42$, $P = 0.002$ for (B). $n = 8$ in each group. **(C)** NpHR and eYFP SNI mice develop significant antinociceptive tolerance to repetitive administration of 10 mg/kg morphine (i.p. twice daily for 6 days). NpHR vs. eYFP: $F_{(5, 70)} = 2.24$, $P = 0.06$; Mor: $i_{(3.14, 43.91)} = 38.95$, $P < 0.0001$; $n = 8$ in each group. **(D)** The effect of yellow light on mechanical PWT on day 7, 30 min after morphine administration in SNI mice. $t = 5.72$, $P < 0.0001$, $n = 8$ in each group. **$P < 0.01$; ns not significant. Two-way repeated measures ANOVAs with Tukey's post-hoc analysis for (A−C); two-tailed unpaired $t$ test for (D). Data are available in S1 Data as a part of Supporting information.
(TIF)

**S10 Fig. Chronic morphine treatment enhances the RE-mPFC projection. (A)** Diagram showing virus injection strategy for labeling of mPFC GABA neurons and mPFC-vlPAG projection neurons and patch-clamp recording of their response to optogenetic activation of RE neurons. Nissl (left part of slices) and anatomical annotations (right part of slices) from the Allen Mouse Brain Atlas (https://mouse.brain-map.org) and Allen Reference Atlas-Mouse Brain (https://atlas.brain-map.org). **(B)** Experimental timeline. **(C−E)** Representative traces (C) and summary of eEPSC amplitude (D, $t = 2.39$, $P = 0.028$) and PPR (E, $t = 1.56$, $P = 0.13$)) recorded from mPFC-vlPAG neurons in response to 20 Hz blue light stimuli (paired pulses) in SNI mice. $n = 10$ cells from 4 mice each group. **(F−H)** Representative traces (F) and summary of eEPSC amplitude (G, $t = 2.15$, $P = 0.045$) and PPR (H, $t = 2.03$, $P = 0.056$) recorded from mPFC GABA neurons in response to 20 Hz blue light stimuli (paired pulses) in SNI mice. $n = 10$ cells from 4 mice each group. *$P < 0.05$, ns not significant. Two-tailed unpaired $t$-tests for (D, E, G, H). Data are available in S1 Data as a part of Supporting information.
(TIF)

**S1 Data. Source data for main figures and supporting figures.**
(XLSX)

## Acknowledgments

We thank Terrence Xiao for data organization, analysis, and grammar check.

## Author contributions

**Conceptualization:** Jun-Li Cao, Cheng Xiao, Chunyi Zhou.

**Data curation:** Shu-Ting Bao, Fang Rao, Cheng Xiao, Chunyi Zhou.

**Formal analysis:** Shu-Ting Bao, Fang Rao, Chunyi Zhou.

**Funding acquisition:** Jun-Li Cao, Cheng Xiao, Chunyi Zhou.

**Investigation:** Shu-Ting Bao, Fang Rao, Cui Yin, Yong Niu, Chunyi Zhou.

**Methodology:** Shu-Ting Bao, Fang Rao, Cui Yin, Yong Niu, Chunyi Zhou.

**Project administration:** Cheng Xiao, Chunyi Zhou.

**Resources:** Cui Yin, Yong Niu, Cheng Xiao, Chunyi Zhou.

**Supervision:** Jun-Li Cao, Cheng Xiao, Chunyi Zhou.

**Validation:** Cui Yin, Yong Niu, Cheng Xiao, Chunyi Zhou.

**Visualization:** Cheng Xiao, Chunyi Zhou.

**Writing – original draft:** Shu-Ting Bao, Cheng Xiao, Chunyi Zhou.

**Writing – review & editing:** Jun-Li Cao, Cheng Xiao, Chunyi Zhou.

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
