## [Editor Report · Decision Letter 0]

3 Dec 2024

Dear Dr Xiao, 

Thank you for submitting your manuscript entitled "Excitatory projections from the nucleus reuniens to the medial prefrontal cortex modulate pain and depression-like behaviors in male mice" for consideration as a Research Article by PLOS Biology.

Your manuscript has now been evaluated by the PLOS Biology editorial staff as well as by an academic editor with relevant expertise, and I am writing to let you know that we would like to send your submission out for external peer review.

Once your full submission is complete, your paper will undergo a series of checks in preparation for peer review. After your manuscript has passed the checks it will be sent out for review. To provide the metadata for your submission, please Login to Editorial Manager (https://www.editorialmanager.com/pbiology) within two working days, i.e. by Dec 05 2024 11:59PM.

Kind regards,

Taylor

Taylor Hart, PhD, 

Associate Editor

PLOS Biology

thart@plos.org

---

## [Decision Letter · Decision Letter 1]

10 Jan 2025

Dear Dr Xiao,

Thank you for your patience while your manuscript "Excitatory projections from the nucleus reuniens to the medial prefrontal cortex modulate pain and depression-like behaviors in male mice" was peer-reviewed at PLOS Biology. It has now been evaluated by the PLOS Biology editors, an Academic Editor with relevant expertise, and by several independent reviewers. 

In light of the reviews, which you will find at the end of this email, we would like to invite you to revise the work to thoroughly address the reviewers' reports.

As you will see below, the reviewers think that the study is well-designed and generally well supported, and complimented the technical scope and level of interest. However, R1 and R3 pointed out missing controls and analyses which will need to be addressed to strengthen the support for some of the conclusions. Both R1 and R3 also recommend repeating the experiment with opioids in the presence of naloxone to further strengthen the study. R2 pointed out the lack of experiments in female mice as a significant limitation of the study, and we strongly encourage repeating some key experiments in female mice as part of a revised submission. The revised submission will also need to make textual changes to address the other concerns of the reviewers.

Given the extent of revision needed, we cannot make a decision about publication until we have seen the revised manuscript and your response to the reviewers' comments. Your revised manuscript is likely to be sent for further evaluation by all or a subset of the reviewers.

**IMPORTANT - SUBMITTING YOUR REVISION**

*Re-submission Checklist*

*Published Peer Review*

*PLOS Data Policy*

*Blot and Gel Data Policy*

Sincerely,

Taylor Hart

Taylor Hart, PhD, 

Associate Editor

PLOS Biology

thart@plos.org

REVIEWS:

Reviewer #1: Using a technical tour de force that includes selective expression of excitatory and inhibitory opsins, rabies-virus based anatomical tracing, in vivo calcium imaging and numerous behavioral tests, the authors establish a role of the thalamic nucleus reunines (RE) in both the sensory and cognitive components of neuropathic pain.

They show that RE neurons respond to mechanical pain stimuli, but not to simple touch, and that Ca signals in RE neurons increases when mice enter the open arms, face an unfamiliar mouse or receive an aversive air puff. They further show that optogenetic activation of RE neurons induces allodynia and aversion in naïve mice, while inhibition of RE neurons alleviates neuropathic pain symptoms in SNI mice. Additionally, the authors demonstrate that the (presumed glutamatergic) RE neurons provide monosynaptic excitatory inputs as well as polysynaptic inhibitory inputs to mPFC neurons that project to the vlPAG. They further analyzed the electrophysiological properties of mPFC-vlPAG neurons in SNI mice and claim that the I:E ratio is increased in these mice compared with sham-operated controls, and that firing of these mPFC neurons from SNI mice cannot reach the same firing frequency as the controls. Finally, the authors probed the behavioral effects of selective activation of the RE inputs to the mPFC. They found that in naïve mice, unilateral optogenetic activation of these inputs causes significant proalgesic effects on the ipsilateral paw, with no effect on the contralateral one. They then proceeded to show that in SNI mice optogenetic inhibition of the RE-mPFC pathway ameliorates both the cognitive/emotional and sensory neuropathic pain symptoms. Finally, they probed the effects of RE-mPFC circuit modulation of the calcium responses of vlPAG neurons elicited by tactile or thermal stimulation and found that inhibition of the RE-PFC pathways potentiates the responses of PAG neurons (ipsilaterally). A final set of experiments show that chemogenetic inhibition of the RE-pPFC pathway potentiates the analgesic response to submaximal doses of morphine. Importantly, the analgesic effect was detectable even in SNI mice with morphine tolerance.

Overall, the results are potentially very interesting, and the multi-pronged technical approach is remarkable. The paper however is not always rigorous, and some conclusions are not fully supported by the data.

Major concerns:

It is unclear how the PWT in SNI mice is ~ 0.01 g in control (off) condition in F 3 E, and then goes to ~0.02 with light on. Yet in Supp. F3, the control value is already higher than 0.02. 

The authors used CamKII promoter to target glutamatergic neurons. Selectivity of this approach should be verified by performing GABA immunostaining because in the cortex CamKII expression is not completely selective for glutamatergic neurons (Veres at a. 2023)

Figure 2K; contrary to what reported in the results, the ChR2 mice do not seem to show any preference on test day. 

Fig 4G. The peak of the inhibitory response is higher in SNI compared with sham and it is concluded that "I:E ratios in mPFC-vlPAG neurons in SNI mice were significantly greater…". However, the decay of the inhibitory current seems faster. Is the charge transfer larger in cells from SNI compared with sham mice? If not, it is very difficult to state that inhibition is increased. In general, I believe that it is not correct to compare the strength of synaptic inputs only on the basis of the current peak. The charge transfer must be considered, at least changes in the current time course cannot be excluded.

The authors assume that cells lacking a voltage sag are interneurons; this however is not a suitable criterion o define interneurons in the mPFC, as commissural projecting pyramidal cells also do not have sag (in both rats and mice; see Dembrow and Johnston 2010; Avesar and Gulledge 2012; Leyre-Jackson and Thomas 2019). Thus, to claim that these recordings were performed from "electrophysiologically identified interneurons" (lines 311-12) is not correct.

Fig 4K-M were these recordings performed in the presence or in the absence of synaptic blockers? 

In the methods (Line 737) it is reported that EPSCs were recorded at -80 mV; in the Figure legend (Line 275) it is reported that they were recorded at -70 mV. Which is true?

The conclusion (Lines 335-38) that "Therefore, in SNI mice, the monosynaptic excitatory effect of the RE-mPFC projection onto mPFC vlPAG neurons may be compromised by enhancement of that onto mPFC GABAergic interneurons innervating mPFC-vlPAG neurons" is not sufficiently supported by the data. The change in maximum firing frequency may be due to intrinsic properties of the cells, not to altered synaptic input. This is easily tested by assessing the I/O curve in the presence of blockers of fast synaptic transmission. 

One important experiment that is missing and, in my opinion, would greatly improve the significance of this paper concerns the potential effects of naloxone on the RE-mPFC pathway dependent analgesia.

Minor points

The timing of each experiment (E.G. FST, TST) relative to SNI/Sham surgery is unclear. Were all tests performed 4 weeks after surgery?

Page 10: It is unclear how animals that have hyperalgesia showed "deficits in pain sensation"

Line 498-500 "Inhibiting the RE-mPFC Glu pathway amplified the pain responses…and ameliorated hyperalgesia" This sentence is self-contradictory

Reviewer #2: This is a nice paper which extends out understanding of brain circuits involved in neuropathic pain states. Nice optogenetics and behavioral measurements were done. Well written, overall well done. 

Comments:

One shortcoming is that only male mice were used in this study. The authors should perform one or two key sets of experiments in female mice 

Line 215: "In ChR2 mice subjected to SNI surgery, optogenetic activation of RE Glu neurons did not further decrease mechanical and thermal hypersensitivity (S3 Fig A, B)". Do you mean "increase hypersensitivity"? Also, this sentence should be moved until after the data on the inhibitory opsin are described. Maybe at line 245

Line 533 - I do not think that there is a contradiction at all. It makes perfect sense that id there is feed forward inhibition in the mPFC, that stimulating inputs from the RE region would lead to less output from the mPFC and thus less excitability in the vlPAG. In fact, the authors already cite papers by Zhang et al (Cell Rep) and Huang et al (Nat Neurosci) - these papers parallel what the authors report here (except for BLA rather than RE inputs). It might be useful to more explicitly discuss the findings in the Huang study (ref 8) vis a vis what the authors observed in their current study - the mechanism sees quite similar especially when it comes to the synaptic processing in the mPFC GABA and pyramidal cells! 

Methods: Viral injections were presumably done on the right hemisphere? If so, please state.

Merthods: SNI surgery - which side? Left? 

Reviewer #3: In this manuscript, Bao et al. investigates the role of the nucleus reuniens (RE) and its projections to the medial prefrontal cortex (mPFC) in modulating neuropathic pain and depression-like behaviors. The authors reveal that RE neurons are activated in response to pain-, anxiety-, and aversion-like stimuli, and the activation is more pronounced in a neuropathic pain state. Activation of the RE-mPFC pathway induces hyperalgesia and depression-like behaviors, while inhibition of this pathway alleviates these behaviors in neuropathic pain mice. Mechanistically, in neuropathic pain, the RE-mPFC pathway preferentially innervates local GABAergic circuits in the mPFC, reducing activity in mPFC-vlPAG neurons. These findings highlight the RE-mPFC pathway as a critical circuit in neuropathic pain and its comorbid depression, and a potential therapeutic target for non-opioid treatments. Overall, in this study, the experiments are reasonably designed, data are solid to support the conclusions, and results are appropriately interpreted. The authors should address my concerns to improve this manuscript.

Main points:

1. In Figure 1I-O, the authors only show the responses of RE neurons to anxiety, social stimulus, and air puff in SNI mice. They should examine these responses in naïve mice. The comparison of the responses between naïve and SNI mice could address whether RE neurons exhibit hyperexcitability and respond differently to these stimuli in SNI mice

2. In Figure 2K, L, the authors should examine movement velocity in two compartments of the CPP chamber during precondition and test sessions. Similar changes should be made in Figure 3 and 5.

3. Figure 4 shows that in mPFC-vlPAG neurons, stimulation of the RE-mPFC glutamatergic projection evoked not only direct EPSCs, but also indirect IPSCs mediated by mFPC GABAergic interneurons. The optogenetic stimulation of the RE-mPFC projection causes less firing in mPFC-vlPAG neurons in SNI mice than sham mice. The authors explained that this phenomenon may be a result of the enhanced indirect IPSCs on mPFC-vlPAG neurons. To confirm this hypothesis, the authors should examine whether the difference in photo-stimulation of the RE-mPFC projection-induced firing between sham and SNI mice is eliminated by inhibition of GABAergic synaptic transmission.

4. Figure 6 shows that inhibition of the RE-mPFC projection enhanced pain responses in vlPAG neurons upon stimulation on the ipsilateral hind paw but not that on the contralateral hind paw. The authors should discuss this interesting result.

5. Figure 7 shows that inhibition of RE neurons projecting to the mPFC exerted analgesic effect in mice subjected to both acute and chronic morphine. The authors should examine whether naloxone can block this analgesic effect to confirm its non-opioid property. Additionally, the authors should clarify whether repetitive morphine administration modifies the RE-mPFC pathway. If this is the case, the role of this pathway will be extended from neuropathic pain to morphine tolerance.

6. S5 Figure shows that mPFC GABAergic interneurons receive innervations from the RE-mPFC projection. The authors should examine whether EPSCs on mPFC GABAergic interneurons induced by optogenetic stimulation of this projection are enhanced in SNI mice. The result can provide further evidence to elaborate the RE-mPFC circuit involved in neuropathic pain.

Minor points

1. In Abstract, the authors should clarify the sentence “the RE-mPFC projection was enhanced with a marked preference for the part innervating local GABAergic circuits to that controlling mPFC neurons projecting to the ventrolateral periaqueductal gray (vlPAG)” . The word “local” is confusing (it means in the RE or mPFC?). The authors should mention “GABAergic circuit in the mPFC”.

2. In Figure 3E-H, the authors should change “NpHR” and “eYFP” into “light”, and add an annotation (NpHR mice or eYFP mice) above each histogram.

3. In line 540-542, “we did observe that optogenetic stimulation of the RE-mPFC projection evoked larger IPSC in SNI mice than in sham mice, resulting in a larger I/E ratio.” Should be changed into “we did observe that optogenetic stimulation of the RE-mPFC projection evoked larger IPSC in mPFC-vlPAG neurons in SNI mice than in sham mice, resulting in a larger I/E ratio.”

---

## [Editor Report · Decision Letter 2]

9 Apr 2025

Dear Dr Xiao,

Thank you for your patience while we considered your revised manuscript "Excitatory projections from the nucleus reuniens to the medial prefrontal cortex modulate pain and depression-like behaviors in male mice" for publication as a Research Article at PLOS Biology. This revised version of your manuscript has been evaluated by the PLOS Biology editors and the Academic Editor.

Based on our Academic Editor's assessment of your revision, we are likely to accept this manuscript for publication, provided you satisfactorily address the following data and other policy-related requests:

- ABSTRACT

It appears that the text of your abstract is duplicated in the submission form. Please check this and ensure that your abstract appears correctly.

- TITLE

Since you have now included data from female mice, we suggest that you modify the title as follows (omitting the word 'male' from the original title): "Excitatory projections from the nucleus reuniens to the medial prefrontal cortex modulate pain and depression-like behaviors in mice". Please update this in the manuscript text and in the submission form.

- FIGURES

Fig. 1D lacks an X-axis label. Please add one (including units). There also appear to be some inconsistencies between the data included in the excel file and the figure. For example, Fig. 1K lists three columns "Naive", "SNI", and "eYFP", but in the excel file, the columns are listed as "eYFP", "Open" and "Close". The data contained in excel column "eYFP" appear to match the figure column labeled as "naive", while data column "open" matches figure column "SNI". But the data in column "close" do not appear to match any of the columns shown on the figure, as the values are all negative while the figure mean for these data appears to be about equal to 1. Please address these issues and double check the other panels so that similar mismatches do not persist elsewhere, and also ensure that all data are displayed properly in the figures.

- FIGURE LEGENDS

Please check that figure elements are explained in the legends. Please also include sample sizes in the figure legends consistently - for example, the sample sizes are listed for Fig. 1 I-K but are missing for Fig. 1 D-H. Please also ensure that figure legends in your manuscript include information on where the underlying data can be found (for example, state in the figure legends that data are available in S1 Data).

- DATA

Thank you for including the source data in the form of a supplementary excel file. This fulfills our requirement that all data be made available without restriction (http://journals.plos.org/plosbiology/s/data-availability). But please make sure that the data match to the figures as mentioned in the previous points. Please check the following points:

Please change the Data Availability Statement in the text to accurately state where the source data are to be found (for example, in S1 Data).

Supplementary files (e.g., excel). Please ensure that all data files are uploaded as 'Supporting Information' and are invariably referred to (in the manuscript, figure legends, and the Description field when uploading your files) using the following format verbatim: S1 Data, S2 Data, etc. Multiple panels of a single or even several figures can be included as multiple sheets in one excel file that is saved using exactly the following convention: S1_Data.xlsx (using an underscore).

- CODE POLICY

We expect to receive your revised manuscript within two weeks.

*Published Peer Review History*

*Press*

Sincerely,

Taylor

Taylor Hart, PhD

Associate Editor

thart@plos.org

PLOS Biology

---

## [Editor Report · Decision Letter 3]

17 Apr 2025

Dear Dr Xiao,

Thank you for the submission of your revised Research Article "Excitatory projections from the nucleus reuniens to the medial prefrontal cortex modulate pain and depression-like behaviors in mice" for publication in PLOS Biology. On behalf of my colleagues and the Academic Editor, Guang Yang, I am pleased to say that we can in principle accept your manuscript for publication, provided you address any remaining formatting and reporting issues. These will be detailed in an email you should receive within 2-3 business days from our colleagues in the journal operations team; no action is required from you until then. Please note that we will not be able to formally accept your manuscript and schedule it for publication until you have completed any requested changes.

PRESS

Sincerely, 

Taylor Hart, PhD, 

Associate Editor

PLOS Biology

thart@plos.org